# Symbolic Governing Equation Discovery Using Neural Arithmetic Modules

## Abstract

Neural architectures with arithmetic inductive biases, such as Neural Arithmetic Logic Units (NALU) and Neural Power Units (NPU), are designed to model arithmetic relationships for improved out-of-distribution extrapolation and interpretability. However, in practice, these models frequently exhibit unstable optimization behaviours such as gradient starvation and convergence to dense and numerically fragile parameterizations that obscure the underlying data structure. We show that arithmetic inductive bias alone is insufficient to guarantee the recovery of sparse symbolic equations. Instead, interpretability should be explicitly enforced through strict architectural constraints. We propose MSRNet, a structured neural framework for extracting sparse symbolic expressions from high-dimensional data. The model has two variants: **MSRNet (Multiplicative Symbolic Regression Network)**, which allows multiplicative and discrete exponential arithmetic interactions via differentiable softmax relaxations, and **ExMSRNet (Extended MSRNet)**, which further allows for logarithmic and exponential pathways. We use a composite training objective that utilizes description-length regularization via entropy-based measures to bias the model towards confident discrete operator selection. Our experiments suggest that MSRNet variants significantly reduce gradient starvation. This could be attributed to explicit constraining of the hypothesis space. We benchmark MSRNet variants on synthetic datasets, SRBench 2025, and AI Feynman I/II/III, where it achieves strong performance with significantly lower computational cost than other symbolic regression methods. Source code for MSRNet is available at: https://anonymous.4open.science/r/MSRNet-6B05

## 1 Introduction

Deep neural networks have achieved remarkable performance across a wide range of domains. However, their internal representations are often uninterpretable, which makes it difficult to understand or verify the reasoning behind their predictions. These models often behave as black boxes. While post-hoc explanation methods provide insights into specific predictions, they do not alter the underlying model to be inherently interpretable (Guidotti et al., 2018; Barredo Arrieta et al., 2020). This lack of interpretability poses a significant challenge in scientific discovery, safety-critical systems, and regulated applications, where transparency and structural understanding are essential (Zhang et al., 2021).

Interpretability in machine learning is broadly concerned with identifying structures and explanations within models. Symbolic regression facilitates interpretability by recovering sparse analytic expressions that describe the data. This approach enables explicit reasoning about learned relationships, rather than relying on feature importance measures and surrogate approximations (Kim et al., 2021). However, symbolic regression methods often scale poorly with dimensionality, and integrating symbolic discovery with neural training remains a fundamental challenge.

Traditional large scale machine learning models lack the ability to model complex non-linear relations. Conversely, modern deep learning models excel at representing complex functions, but typically rely on dense, entangled parameterizations that hinder semantic interpretation and performs poorly on out-of-distribution data. To bridge this gap, recent work has introduced neural arithmetic modules, such as the Neural Ac-

cumulator (NAC) and Neural Arithmetic Logic Unit (NALU), which incorporates strong inductive biases toward learning arithmetic operations, specifically addition and multiplication (Trask et al., 2018; Madsen & Johansen, 2020). This approach allows the model to generalize outside the training range, which is critical for scientific applications, where experimental data are generally scarce. These models allow for a more faithful representation of structured, symbolic-like computations within a neural network framework than generic multi-layer perceptrons.

However, our results suggest that arithmetic inductive bias alone is insufficient to guarantee interpretability. Prior analyses have shown that NALU and NPU models can exhibit unstable optimization behaviour, sensitivity to input scale, and convergence to dense or numerically fragile parameterizations, often resulting in solutions that are functionally correct but structurally uninterpretable (Schlör et al., 2020; Madsen & Johansen, 2020; Mistry et al., 2022). In practice, Neural Arithmetic Logic Modules (NALMs) frequently fail to converge to clean symbolic solutions and instead exhibit unstable behaviour. This failure is primarily driven by two phenomena:

- Gradient Starvation: Dominant high-magnitude terms receive most of the gradient signal, while weaker but semantically essential terms are suppressed during optimization (Pezeshki et al., 2021). This issue is especially pronounced in NALMs, where exponential operations often generate disproportionately large values (Schlör et al., 2020; Madsen & Johansen, 2020; Mistry et al., 2022).

- Representational Ambiguity and Division Singularity: Continuous parameterizations result in large equivalence classes where multiple dense, compensatory weight configurations yield identical functional behaviour (Zhao et al., 2026). Furthermore, inverse relationships introduce a "Division Singularity" where gradients approach infinity near zero, causing optimizers to suppress division pathways entirely and fail to discover inverse laws (Mistry et al., 2021; Madsen & Johansen, 2020).

Since standard optimization objectives do not encode interpretability, and multiple functionally equivalent yet structurally distinct solutions exist, interpretability cannot be assumed to arise from inductive bias alone (D'Amour et al., 2022; Fort et al., 2019; Locatello et al., 2019; Lipton, 2018). Instead, it requires explicit architectural or objective-level constraints (Rudin, 2019; Cranmer et al., 2020; Plumb et al., 2020). We propose Multiplicative Symbolic Regression Networks (MSRNet), a structured arithmetic neural framework in which both additive and multiplicative interactions are selected from discrete operator sets via softmax function, coupled with regularization terms that bias the model toward compact, confident arithmetic representations. By constraining both the selection of input variables and the form of arithmetic interactions, the model is encouraged to learn sparse, disentangled representations that allow for a direct symbolic interpretation. We use RealNPU with discrete parameterizations as the core for non-linear interaction modelling. Unlike traditional sparse regression approaches over fixed libraries (Brunton et al., 2016; Rudy et al., 2017; McConaghy, 2011), our method embeds operator selection within a differentiable neural architecture, enabling end-to-end learning of arithmetic structures. This approach combines the expressiveness of neural networks with the structural clarity of symbolic systems. Experimental results demonstrate that MSRNet variants can recover meaningful symbolic-like expressions with minimal loss in prediction accuracy.

## 2 Related Works

### 2.1 Interpretability in Machine Learning

Interpretability in machine learning is an active and diverse research area motivated by the need for transparency, trust, and explainability in predictive models (Guidotti et al., 2018; Ribeiro et al., 2016; Marcinkevičs & Vogt, 2023). Early work emphasizes the need for a rigorous and context-dependent definition of interpretability, highlighting that transparency, simulatability, and decomposability are distinct and often competing objectives (Lipton, 2018; Doshi-Velez & Kim, 2017).

Recent surveys have formalized these methods into a taxonomy distinguishing between *post-hoc* explanation techniques that analyze fixed black-box models, and *intrinsically interpretable* models whose structure is designed to be identifiable (Doshi-Velez & Kim, 2017; Molnar, 2025). Post-hoc explanation methods, such as

feature attribution and surrogate modeling, aim to explain the behaviour of black-box models after training (Ribeiro et al., 2016; Lundberg & Lee, 2017). While effective in some settings, these approaches do not alter the underlying model structure and may fail to faithfully represent the true decision process (Lipton, 2018). Consequently, for high-stakes scientific discovery, recent literatures increasingly emphasises that we must "stop explaining black boxes" and instead focus on architectures that are intrinsically interpretable, ensuring that the model's structure itself serves as the explanation (Rudin, 2019).

## 2.2 Symbolic Regression and Equation Discovery

Symbolic regression is a class of methods that aims to identify closed-form human-readable mathematical expressions that model the observed data (Schmidt & Lipson, 2009; Petersen et al., 2019; Cranmer et al., 2020). Traditional approaches typically rely on evolutionary strategies or combinatorial search over expression spaces, optimizing for both prediction accuracy and expression complexity (Koza, 1992; Vladislavleva et al., 2008; Smits & Kotanchek, 2005). Recent surveys suggest that symbolic regression is emerging as a promising machine learning method for inferring succinct mathematical forms directly from data. It has applications in various fields including science and engineering where interpretable models are essential (Makke & Chawla, 2024; Aldeia & de França, 2022). Despite its interpretability advantages, Genetic Programming based methods notoriously suffer from "bloat", i.e., the uncontrolled growth of expression complexity without corresponding gains in accuracy, which hampers their scalability to high-dimensional problems (Silva & Costa, 2009). Furthermore, genetic programming is inherently discrete and non-differentiable, which makes it difficult to integrate with modern deep learning pipelines (Petersen et al., 2019; Mundhenk et al., 2021). This has motivated a shift towards Neural Symbolic Regression, where differentiable architectures allow for gradient-based optimization of symbolic structures, combining the expressivity and efficiency of neural networks with the conciseness of analytic equations (Martius & Lampert, 2016a; Kim et al., 2021; Biggio et al., 2021; Tohme et al., 2024).

Sparse-library methods and physics-inspired systems such as AI Feynman provide strong symbolic recovery in many settings, but they often rely on some explicit candidate-library (Udrescu & Tegmark, 2020; Udrescu et al., 2020; Brunton et al., 2016). In contrast, neural approaches offer flexibility and scalability, but they generally do not impose explicit mechanisms to enforce structural sparsity or yield compact symbolic forms (d'Avila Garcez et al., 2019; Cranmer et al., 2020; Udrescu & Tegmark, 2020). Furthermore, their inherent complexity poses interpretability challenges that require separate post hoc explanation methods (Ribeiro et al., 2016; Lundberg & Lee, 2017), an approach often criticized in high-stakes domains (Rudin, 2019). This work aims to bridge this gap by incorporating a sparse arithmetic structure within a neural framework.

Parallely, Standardized evaluation has become central to progress in symbolic regression. SRBench (La Cava et al., 2021) provides a common benchmark setting for cross-method comparison by testing methods against hundreds of real-world tabular datasets and known physical equations. SRBench 2025 (Imai Aldeia et al., 2025) extends this benchmarking by moving away from simple aggregated rankings. Instead, they emphasize problem-specific topologies and multi-objective trade-offs, capturing the balance between accuracy, expression complexity, and computational cost. Separating the evaluation into distinct tracks allows us to independently measure two different capabilities: The black-box track evaluates a model's ability to maximize predictive accuracy and generalize on noisy, real-world data, while the fundamental-equation track tests whether a method can recover the exact underlying ground-truth analytic expression. There have been efforts to incorporate domain-expert feedback to measure actual scientific utility beyond simple length penalties (de Franca et al., 2024).

## 2.3 Neural and Neuro-Symbolic Approaches to Interpretable Modeling

Neural network architectures tailored for interpretability often combine principles from symbolic regression with the representational capability of deep learning (Kim et al., 2021). For instance, Equation Learner (EQL) networks are designed to embed symbolic operations such as multiplication into neural layer activation functions, enabling the model to learn analytic functional forms end-to-end via backpropagation (Martius & Lampert, 2016b; Sahoo et al., 2018).

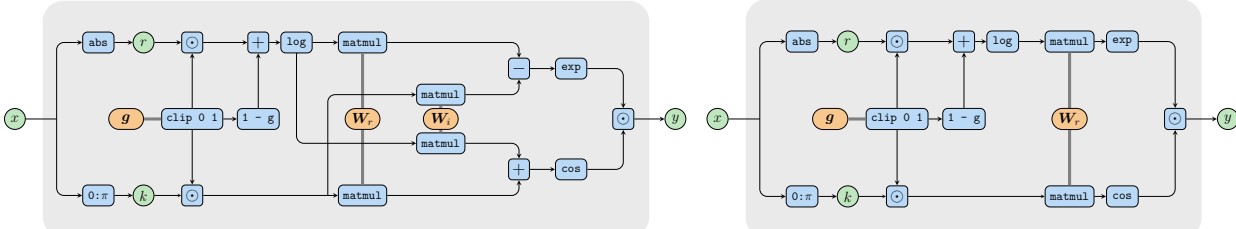

Figure 1: NPU (left) and RealNPU (right) computation graphs (Heim et al., 2020).

The intersection of symbolic regression and neural modeling reflects a broader interest in neuro-symbolic integration, where structured symbolic components are embedded within neural architectures to balance expressivity with interpretability (Besold et al., 2017; Mao et al., 2019). This includes universal tools that combine neural fitting with symbolic inference to recover governing equations from data in scientific domains (Hu et al., 2024). These hybrid approaches highlight a key challenge of designing mechanisms that combine continuous optimization with discrete symbolic structure in a way that is both trainable and semantically meaningful (Raedt et al., 2020).

The intersection of symbolic reasoning and neural learning is often characterized as the "third wave" of AI (Garcez & Lamb, 2023; Launchbury, 2017). Colelough & Regli (2025) highlight that while significant progress has been made in learning and inference, explainability and trustworthiness remain under-explored, particularly in continuous domains. Our approach follows this direction by constraining neural learning to remain symbolically compatible during training, instead of relying only on post-hoc explanations.

## 2.4 Neural Arithmetic Modules

Neural arithmetic modules were designed to address the inability of standard neural networks to extrapolate arithmetic relations beyond the training distribution. The Neural Arithmetic Logic Unit (NALU) imposes constraints on their parameters to encourage learning addition, subtraction, multiplication, and division (Trask et al., 2018). These models achieve improved generalization on synthetic arithmetic tasks compared to conventional architectures. However, subsequent analyses revealed several limitations of these modules, notably sensitivity to initialization, instability during training, and poor recovery of clean, sparse representations in practice (Schlör et al., 2020; Madsen & Johansen, 2020; Mistry et al., 2021). Further improvements in these modules were aimed at improving training stability (Schlör et al., 2020). Madsen & Johansen (2020) introduced Neural Addition Units (NAU) and Neural Multiplication Units (NMU), which remove the gating mechanism in favor of direct linear and multiplicative operations. Neural Power Units (NPU) extend this family by modeling exponentiation in log space using magnitude-phase decompositions implemented via logarithmic and trigonometric operations, increasing expressiveness (Heim et al., 2020).

We empirically observe that correct functional behaviour does not necessarily correspond to interpretable parameter values, particularly in the presence of competing terms and high-dimensional inputs. Further, multiplicative units remain prone to "Gradient Starvation", a phenomenon where the optimization captures only dominant features (high magnitude terms) while starving weaker, yet semantically relevant, signals (Pezeshki et al., 2021). These observations motivate the need for additional structural constraints beyond arithmetic inductive bias alone.

# 3 Methodology

This section describes MSRNet, our arithmetic neural framework for symbolic equation discovery. The central principle is to enforce interpretability through architectural constraints.

## 3.1 Architecture Overview

MSRNet is organized into following two stages:

1. **Discrete additive feature selection:** We use an NAC module with discrete weights, hereafter referred to as the Discrete NAC (or DNAC), and

2. **Discrete RealNPU arithmetic core:** We model multiplicative and exponential interactions using a RealNPU module with discrete rational weights.

This modular decomposition expands the hypothesis space at each stage, while maintaining symbolic representational compatibility and preserving end-to-end differentiability.

## 3.2 Discrete Additive Feature Selection

The first stage performs sparse linear interaction using a discrete Neural Accumulator (NAC) parameterization (Trask et al., 2018). Given $\mathbf{x} \in \mathbb{R}^d$, the transformed features are

$$\tilde{x}_i = \sum_{j=1}^{d} a_{ij} x_j, \tag{1}$$

where

$$a_{ij} \in \{-1, 0, 1\}. \tag{2}$$

To retain differentiability during training, coefficients are relaxed by a softmax parameterization:

$$a_{ij} = \sum_{k \in \{-1,0,1\}} k \cdot p_{ij}(k), \qquad p_{ij}(k) = \mathrm{softmax}(\theta_{ij})_k. \tag{3}$$

This formulation enforces exact sign control and explicit feature exclusion when $a_{ij} = 0$, yielding semantically interpretable sparse additive structure. This stage outputs an $h$ dimensional vector $\tilde{\mathbf{x}} \in \mathbb{R}^h$. We call this $h$, the hidden dimension.

## 3.3 Restricted Arithmetic Expansion

For modelling multiplicative and exponential interactions, we use a RealNPU module with discrete parameterizations (Heim et al., 2020). To make the symbolic expression extraction from model easier and interpretable, our implementation is strictly real-valued and uses no imaginary weights. For transformed features $\tilde{\mathbf{x}}$, we define

$$\mathbf{h}(\tilde{\mathbf{x}}) = [h_1(\tilde{\mathbf{x}}), \ldots, h_m(\tilde{\mathbf{x}})]^\top, \tag{4}$$

where

$$h_i(\tilde{\mathbf{x}}) = \prod_j \tilde{x}_j^{w_{ij}}. \tag{5}$$

Each exponent $w$ is selected from a fixed vocabulary of interpretable rational values

$$\mathcal{W} = \left\{ -3, -2, -1, -\tfrac{1}{2}, -\tfrac{1}{3}, 0, \tfrac{1}{3}, \tfrac{1}{2}, 1, 2, 3 \right\}. \tag{6}$$

Exponent selection is relaxed through

$$w_{ij} = \sum_{k \in \mathcal{W}} k \cdot p_{ij}(k). \tag{7}$$

We choose these values to cover common arithmetic structure found in physics while keeping the model identifiable: $\{\pm 1, \pm 2, \pm 3\}$ captures linear, polynomial, and inverse-power behaviour, $\{\pm \tfrac{1}{2}, \pm \tfrac{1}{3}\}$ captures root and reciprocal-root relations, and 0 enables explicit feature suppression. Restricting $\mathcal{W}$ to small rational exponents reduces the hypothesis space and suppresses dense compensatory parameterizations. It results in a reduction in representational ambiguity and produces more compact, human-interpretable recovered equations. Exponentiation is computed in log space with a small stability constant $\epsilon$ and sign preserving real-valued transforms for negative inputs via magnitude-phase decomposition.

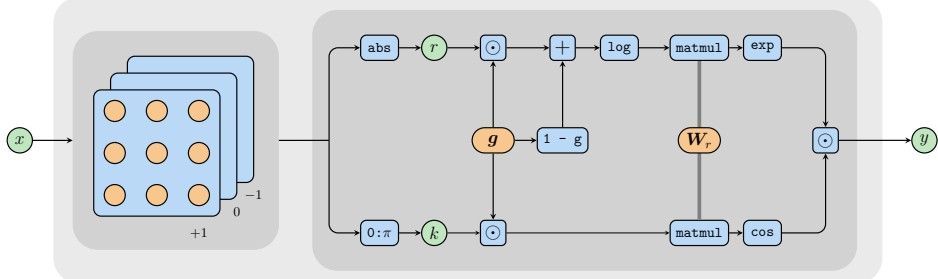

Figure 2: Architectural overview of MSRNet architecture: A discrete NAC feature selector is followed by a discrete RealNPU module to recover compact, expressive equations.

### 3.4 Model Variants

We evaluate two variants built on the same feature-selection and multiplicative core:

- **MSRNet (Multiplicative Symbolic Regression Network):** Uses the RealNPU multiplicative arithmetic module with discrete parameters. The output of this module is:

$$\mathbf{z}_{\text{MSR}} = \exp(W\mathbf{r}) \odot \cos(W\pi\mathbf{k}), \tag{8}$$

  where,

$$\mathbf{r} = \mathbf{g} \odot log\left(|\tilde{\mathbf{x}}| + \epsilon\right) + (1 - \mathbf{g}), \quad \mathbf{k} = \mathbf{g} \odot 1[\tilde{\mathbf{x}} < 0]. \tag{9}$$

  This is equivalent to the multiplicative form defined under RealNPU.

- **ExMSRNet (Extended MSRNet):** Introduces two discrete binary gates ($g_{log}$ and $g_{exp}$) which allows the model to bypass log and/or exp operations for better representational capacity. Let $\mathbf{g}_{\text{log}}$ and $\mathbf{g}_{\text{exp}}$ denote their effective (relaxed) selections. ExMSRNet multiplicative unit computes:

$$\mathbf{a} = \mathbf{g} \odot \tilde{\mathbf{x}} + (1 - \mathbf{g}), \tag{10}$$

$$\mathbf{u} = \mathbf{g}_{\text{log}} \odot \log(|\mathbf{a}| + \epsilon) + (1 - \mathbf{g}_{\text{log}}) \odot \mathbf{a}, \tag{11}$$

$$\mathbf{z} = \mathbf{g}_{\text{exp}} \odot \left(\exp(W\mathbf{u}) \odot \cos(W(\pi\mathbf{k} \odot \mathbf{g}_{\text{log}}))\right) + (1 - \mathbf{g}_{\text{exp}}) \odot \mathbf{u}. \tag{12}$$

  where $\mathbf{k}$ is the gated sign indicator defined as $\mathbf{k} = \mathbf{g} \odot 1[\tilde{\mathbf{x}} < 0]$. The log gate controls whether each channel is processed in log space. The exp gate controls whether the module emits the exponentiated magnitude-phase branch or a passthrough branch.

MSRNet provides a strict arithmetic baseline, while ExMSRNet increases expressivity for nonlinear targets such as exponential and logarithmic functions. When multiple modules are required, we introduce trainable vectors $m_i$ and compose modules as $y = \text{MSRNet}_2(m_1 \odot \text{MSRNet}_1(x))$. This formulation extends to an arbitrary composition depth, with the final MSRNet output passed through a linear layer. With this compositional construction, the model can represent more complex equations such as $x_a^{x_b}$.

### 3.5 Problem Formulation

We consider supervised regression with dataset

$$\mathcal{D} = \left\{ \left(\mathbf{x}^{(i)}, y^{(i)}\right) \right\}_{i=1}^{N}, \qquad \mathbf{x}^{(i)} \in \mathbb{R}^d, \ y^{(i)} \in \mathbb{R}. \tag{13}$$

We assume that the data-generating process admits a compact arithmetic expression over a sparse subset of variables, composed of operations such as addition, multiplication, exponentiation, and logarithms. This assumption is consistent with many real-world physical formulas. The objective is twofold: (i) to minimize prediction error, and (ii) to recover a sparse and structurally identifiable symbolic expression. Any dataset used is standardized before passing it through the model to mitigate extreme magnitudes and gradient starvation during optimization.

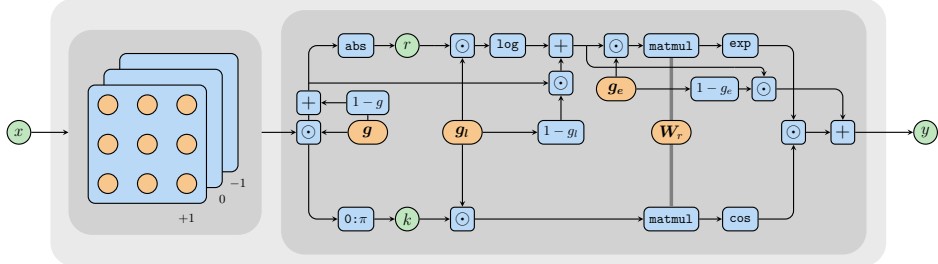

Figure 3: ExMSRNet architecture with explicit binary $g_{log}$ and $g_{exp}$ gating.

## 3.6  Training Objective

To jointly optimize predictive accuracy and symbolic simplicity, we use the following composite objective:

$$\mathcal{L} = \mathcal{L}_{\text{task}} + \lambda_{\text{DL}}\mathcal{L}_{\text{DL}} + \lambda_{\text{order}}\mathcal{L}_{\text{order}} + \lambda_{\text{sparsity}}\mathcal{L}_{\text{sparsity}}, \tag{14}$$

where $\mathcal{L}_{\text{task}}$ is mean squared error and

$$\mathcal{L}_{\text{order}} = \sum_{i,j} \phi(p_{ij}) \tag{15}$$

penalizes high-order interactions using a function $\phi$, which maps the exponential weights to their respective loss value. $\mathcal{L}_{\text{sparsity}}$ applies $\ell_1$ regularization (Tibshirani, 2018) to additive coefficients to encourage sparse features. To stabilize optimization of discrete operator probabilities, we apply temperature annealing to the softmax relaxation during training (Hinton et al., 2015; Jang et al., 2017; Maddison et al., 2017). Specifically, operator distributions are computed as

$$p_{ij}^{(t)}(k) = \text{softmax}\left(\frac{\theta_{ij}(k)}{\tau_t}\right), \qquad \tau_t = \max\left(\tau_{\min}, \tau_0 \alpha^t\right), \tag{16}$$

where $t$ is the training step. A higher initial temperature encourages smoother gradients and exploration, while gradual cooling sharpens selections toward near-discrete operators, and improves final symbolic sparsity and structural confidence.

## 3.7  Description-Length Regularization and Entropy Metric

To favor concise arithmetic equations, we regularize operator distributions with an MDL-inspired entropy term. For each categorical operator distribution $p_i$, we compute

$$H_i = -\sum_k p_i(k) \log p_i(k), \tag{17}$$

and use the aggregate

$$\mathcal{L}_{\text{DL}} = \sum_i H_i, \tag{18}$$

as a training regularizer (Grünwald, 2007). Here, entropy serves two roles in our framework: (i) as a loss term that simulates description-length regularization (Rissanen, 1978), and (ii) as a structural-confidence metric. Low entropy indicates near-deterministic operator selection and high structural confidence, while high entropy indicates ambiguity between competing arithmetic forms.

## 3.8  Symbolic Extraction

After convergence, each operator distribution is collapsed to its maximum-probability choice, and zero-valued coefficients are pruned exactly. The resulting arithmetic expression is then algebraically simplified using python sci-py library (Virtanen et al., 2020). Because symbolic choices are represented during training, this extraction is faithful by construction and does not require secondary fitting or post-hoc surrogate approximation.

# 4 Theoretical Analysis

This section analyzes why unconstrained arithmetic neural units often fail to recover interpretable structure and how discrete operator selection combined with description-length regularization improves optimization stability and identifiability.

## 4.1 Representational Ambiguity in Multiplicative Models

Consider multiplicative arithmetic units of the form

$$h(x) = \prod_{j=1}^{d} x_j^{w_j} \tag{19}$$

This parameterization yields large equivalence classes of representations. For example, for any $\epsilon \in \mathbb{R}$,

$$x = x^{1+\epsilon} \cdot x^{-\epsilon}, \tag{20}$$

yields identical functional behaviour while producing dense, cancelling parameter configurations. Such equivalences prevent reliable recovery of symbolic structure despite low prediction error (Schlör et al., 2020). In high-dimensional settings, this ambiguity grows combinatorially, making identifiability challenging without additional constraints.

By restricting coefficients and exponents to discrete operator sets, the proposed framework reduces representational ambiguity. Discrete selection collapses equivalence classes by eliminating infinitesimal compensations between parameters. In particular, exact zero selections remove entire multiplicative paths, preventing dense equilibria that arise in unconstrained models. This restriction reduces the effective hypothesis space and improves identifiability under low data coverage scenarios. Furthermore, the description-length regularization ($\mathcal{L}_{DL}$) explicitly penalizes high-entropy states, ensuring that the model converges to sparse, confident, and identifiable configurations.

Further, since exponents are selected from a fixed vocabulary $\mathcal{W}$, they cannot drift to unrestricted values during optimization. This contrasts with unconstrained NPU parameterizations where continuous exponent weights may grow to arbitrary magnitudes, often resulting in unstable optimization.

## 4.2 Gradient Starvation in Arithmetic Neural Units

Consider a target function composed of multiple arithmetic terms:

$$y = ax_i^2 + bx_j x_k + cx_\ell. \tag{21}$$

As $|x_i|$ grows large, it asymptotically dominates the function's behaviour. The impact of $x_j$, $x_k$, and especially $x_\ell$ on $y$ reduces as $x_i$ grows in magnitude. In multiplicative architectures, the gradient of the loss with respect to parameters associated with each term scales with the magnitude of that term. If one component dominates numerically, gradients corresponding to weaker but semantically meaningful terms are suppressed. This phenomenon, known as *gradient starvation* (Pezeshki et al., 2021), has been empirically observed in NALU and NPU models (Schlör et al., 2020; Madsen & Johansen, 2020; Heim et al., 2020).

Formally, let $\mathcal{L}$ be a squared-error loss. Then

$$\frac{\partial \mathcal{L}}{\partial w_j} \propto (h(x) - y) \cdot h(x) \cdot \log |x_j| \tag{22}$$

Therefore, terms with larger contribution to $h(x)$ dominate gradient updates. As a result, smaller terms may never be learned even when they are part of the true data-generating process.

### 4.3 Computational Complexity

Let $d$ denote the input dimensionality and $|\mathcal{W}|$ the size of the discrete exponent set. For some hidden dimension $h$, the additive stage introduces $O(d \cdot h)$ parameters, while the multiplicative stage introduces $O(d \cdot h \cdot |\mathcal{W}|)$ parameters. Compared to the unconstrained RealNPU, this increases the parameter count by a factor of $|\mathcal{W}|$, but significantly reduces the effective hypothesis space. The computational overhead of softmax-based operator selection is negligible relative to the cost of standard neural layers, and the model remains fully compatible with stochastic gradient descent.

In contrast, other neuro-symbolic regression methods, such as SINDy (Brunton et al., 2016), rely on explicit expansion with a candidate function library that includes polynomial and interaction terms. For polynomial interactions up to degree $k$, the size of the candidate library scales as

$$\mathcal{O}\left(\sum_{i=0}^{k} \binom{d+i-1}{i}\right), \tag{23}$$

which grows combinatorially with input dimensionality. When cross-terms, higher-order interactions, and exponentiations are included, the library size rapidly becomes intractable even for moderate values of $d$.

The proposed framework avoids explicit input space expansion by implicitly representing arithmetic interactions through a differential neural architecture. Discrete operator selection enables the model to explore a rich space of arithmetic expressions while maintaining parameter growth that is linear in $d$ and $|\mathcal{W}|$. As a result, this approach scales more favorably to higher-dimensional settings than other neuro-symbolic methods, while retaining the ability to recover compact symbolic expressions.

### 4.4 Deceptive Parameters with RealNPU

Even when a RealNPU module achieves low predictive error, its learned parameters can be semantically deceptive. The primary reason is that the real-valued phase-magnitude decomposition does not explicitly model the complex phase. RealNPU suppresses the intermediate imaginary component $i$ that is required for consistent exponentiations on negative inputs.

Consider

$$y = \sqrt{x_a x_b}, \tag{24}$$

with some training data where pairs $(x_a, x_b)$ are of the same sign, either positive or negative. A parameterization that appears correct on observed samples can still output errorneous behaviour under sign changes: For $x_a < 0$ and $x_b < 0$, it would return $-\sqrt{x_a x_b}$ despite having the correct ground truth equation representation. A similar ambiguity appears for odd roots of negative values. For example,

$$(-1)^{1/3} = e^{i\pi/3} = \frac{1}{2} + i\frac{\sqrt{3}}{2}. \tag{25}$$

Therefore, a real-only projection collapses to $1/2$.

This is an inherent limitation of RealNPU parameterization. The complex NPU can represent intermediate complex-phase terms and therefore handles such behaviour more faithfully. While, complex-valued internal state improves functional coverage, it makes direct parameter-level interpretability and symbolic extraction cumbersome.

### 4.5 Limitations of Theoretical Guarantees

The guarantees implied by discrete operator selection and description-length regularization depend on several assumptions. First, sufficient data coverage is required to distinguish competing arithmetic explanations. However, compared to other Neural Arithmetic Modules, MSRNet achieves better equation recovery under low data coverage scenarios. Second, when multiple expressions are functionally equivalent over the observed domain, unique recovery cannot be guaranteed. Finally, the imposed constraints intentionally restrict

expressivity. Functions requiring dense interactions or non-rational exponents lie outside the target hypothesis class. These limitations are inherent to any approach that prioritizes interpretability over universal approximation.

## 5 Experimental Setup

The evaluation protocol is organized into a primary benchmark and two secondary benchmarks. The primary benchmark is a synthetic dataset designed to stress-test feature selection and nonlinear arithmetic recovery. Secondary benchmarks evaluate external validity on established symbolic regression datasets.

### 5.1 Datasets

For each synthetic task, we generate 40,000 samples using

$$y = f(\mathbf{x}) + \eta, \tag{26}$$

where each input dimension is sampled independently from $\mathcal{U}[-5, 5]$, and $\eta \sim \mathcal{N}(0, \sigma^2)$ with small $\sigma$.

The primary tasks are:

- **Large Dimensions** $(d = 100)$: $y = (x_{93} + x_{11})(x_2 + x_{80}) + \mathcal{N}(0, 0.5^2)$,

- **Exp** $(d = 5)$: $y = e^{x_3} + \mathcal{N}(0, 0.5^2)$,

- **Sin** $(d = 5)$: $y = sin(x_3) + \mathcal{N}(0, 0.1^2)$,

- **Lure** $(d = 5)$: $y = x_3 x_4 + 100 x_5 + \mathcal{N}(0, 0.5^2)$.

Large Dimensions tests sparse discovery in high-dimensional noise, while Lure tests robustness to dominant-term distraction.

**SRBench 2025** is used as the secondary benchmark, with two tracks: *black-box* and *fundamental equations* (Imai Aldeia et al., 2025; La Cava et al., 2021). We obtain the SRBench 2025 tasks from the PMLB dataset repository (Olson et al., 2017; Romano et al., 2021).

**AI Feynman I/II/III** is used as an additional equation-recovery benchmark (Udrescu & Tegmark, 2020; Udrescu et al., 2020), with explicit tracking of module count required for each recovered formula. We obtain the AI Feynman datasets from the PMLB dataset repository (Olson et al., 2017; Romano et al., 2021).

### 5.2 Compared Methods

We compare MSRNet and ExMSRNet against the following neural arithmetic baselines:

- **NALU (NAC$_\bullet$)** (Trask et al., 2018),

- **NMU** (Madsen & Johansen, 2020),

- **RealNPU** (Heim et al., 2020),

- **NPU** (Heim et al., 2020).

For NALU and NMU, we apply explicit feature expansion over $\mathcal{W}$ because these models do not natively represent exponentiation. For each of these models, we compare the effect of feature selection using a linear function with $\ell_1$ regularization against DNAC. For SRBench comparisons, we additionally report against the various symbolic-regression baselines.

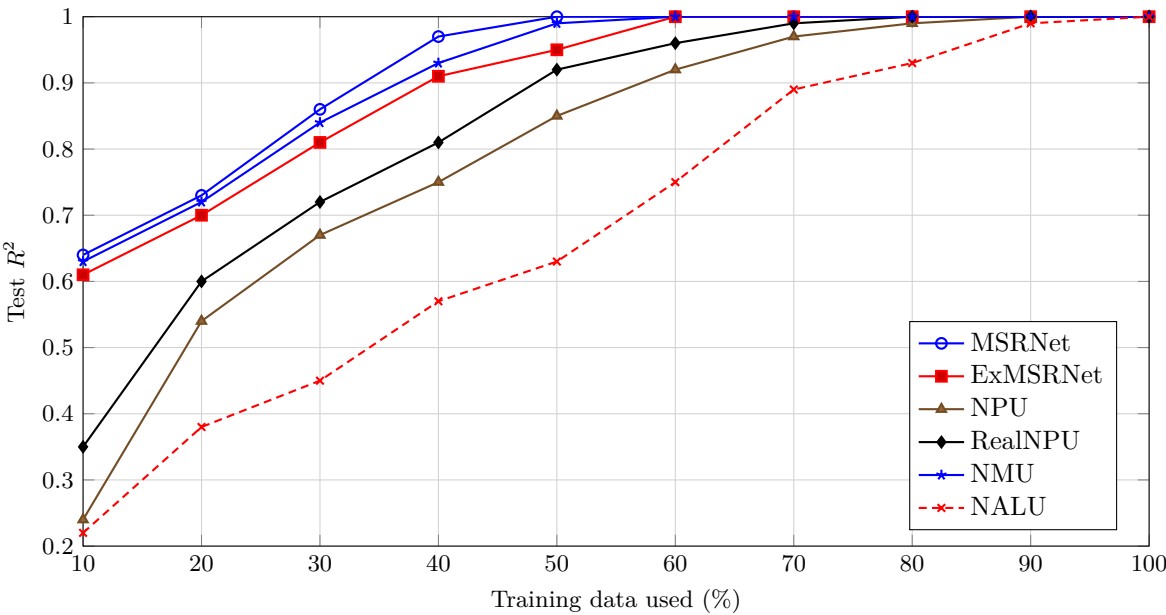

Figure 4: Scarce-data regime: Large Dimensions test $R^2$ versus fraction of training data used on synthetic datasets.

### 5.3 Training Details

Reported metrics are computed on held-out test sets and aggregated across thirty different seeds using median and quartiles. For MSRNet and ExMSRNet, discrete distributions (additive selectors, exponent selectors, and all the gates) use Xavier-Uniform initialization to prevent early collapse (Glorot & Bengio, 2010). For synthetic datasets and AI Feynman, we use a 75/25 train/test split since it's inline with the approach used by SRBench. Further, we use three-fold cross-validation on the training dataset. Regularization strengths for description length, interaction order, and sparsity are selected via grid search over cross-validation splits. Energy and emissions are tracked with `eco2ai` on an NVIDIA Tesla P100 GPU for training (Budennyy et al., 2023; NVIDIA Corporation, 2016).

## 6 Results

### 6.1 Primary Synthetic Benchmark

On the synthetic datasets, DNAC consistently performs better than linear feature selection. This effect is especially prominent in Large Dimensions benchmark, where the feature selector has to select sparse features (4 useful features out of 100). Out of all these methods, only MSRNet, ExMSRNet, and NMU are able to extract the underlying data generating equation correctly. However, NMU and NALU consume roughly two times more memory and energy because feature expansion produces high-dimensional inputs. MSRNet variants consistently outperforms NALU and NPU variants in sin, exp, and lure benchmarks, with ExMSRNet getting perfect test $R^2$ on exp with every run. The gains are largest on Lure, where the NALU and NPU variants suffer from gradient starvation.

Removing DNAC feature selection increases expression density, worsens equation recovery fidelity, and degrades test $R^2$ performance, suggesting that feature gating is a primary contributor to robustness.

Table 1: Synthetic benchmark comparison across methods (hidden dimension = 5). We compare median test $R^2$ [$1^{st}$ quartile, $3^{rd}$ quartile] followed by mean energy usage ($10^-3$ Wh) $\pm$ std on the next line. The best median test $R^2$ score is highlighted in **bold**.

| Method | Feature Selector | Large Dimensions | Exp | Sin | Lure |
|---|---|---|---|---|---|
| NALU | Linear | 0.75[0.68 - 0.84] | 0.95[0.81 - 0.96] | 0.74[0.72 - 0.79] | 0.85[0.76 - 0.87] |
| | | 10.99 $\pm$ 2.10 | 6.24 $\pm$ 1.16 | 6.32 $\pm$ 2.09 | 5.67 $\pm$ 1.05 |
| | DNAC | **1.00[0.99 - 1.00]** | 0.96[0.93 - 0.96] | 0.76[0.74 - 0.81] | 0.85[0.80 - 0.88] |
| | | 10.28 $\pm$ 1.18 | 6.37 $\pm$ 0.96 | 6.33 $\pm$ 1.96 | 5.67 $\pm$ 0.85 |
| NMU | Linear | 0.85[0.84 - 0.88] | 0.99[0.93 - 0.99] | **0.96[0.87 - 0.97]** | **1.00[0.97 - 1.00]** |
| | | 12.05 $\pm$ 1.78 | 7.15 $\pm$ 0.23 | 7.71 $\pm$ 0.74 | 7.84 $\pm$ 1.29 |
| | DNAC | **1.00[0.99 - 1.00]** | 0.99[0.98 - 1.00] | 0.95[0.90 - 0.98] | **1.00[1.00 - 1.00]** |
| | | 12.44 $\pm$ 0.69 | 7.35 $\pm$ 0.41 | 7.72 $\pm$ 0.71 | 7.91 $\pm$ 1.07 |
| RealNPU | Linear | 0.80[0.76 - 0.85] | 0.99[0.92 - 0.99] | 0.91[0.87 - 0.92] | 0.97[0.94 - 0.98] |
| | | 5.77 $\pm$ 2.29 | 10.82 $\pm$ 6.84 | 8.23 $\pm$ 3.27 | 11.59 $\pm$ 5.74 |
| | DNAC | **1.00[0.99 - 1.00]** | 0.99[0.97 - 0.99] | 0.91[0.89 - 0.92] | 0.97[0.94 - 0.98] |
| | | 6.15 $\pm$ 1.78 | 10.83 $\pm$ 6.71 | 8.25 $\pm$ 2.95 | 11.98 $\pm$ 4.83 |
| NPU | Linear | 0.79[0.76 - 0.82] | 0.99[0.87 - 0.99] | **0.96[0.80 - 0.96]** | 0.96[0.89 - 0.97] |
| | | 6.71 $\pm$ 1.53 | 7.79 $\pm$ 1.85 | 15.20 $\pm$ 10.93 | 17.11 $\pm$ 15.22 |
| | DNAC | **1.00[0.99 - 1.00]** | 0.99[0.96 - 0.99] | **0.96[0.93 - 0.97]** | 0.95[0.93 - 0.96] |
| | | 6.93 $\pm$ 0.91 | 7.79 $\pm$ 1.61 | 15.60 $\pm$ 10.90 | 19.71 $\pm$ 13.69 |
| MSRNet | Linear | 0.83[0.80 - 0.85] | 0.98[0.93 - 0.99] | 0.95[0.92 - 0.96] | 0.98[0.95 - 0.99] |
| | | 4.15 $\pm$ 2.32 | 4.22 $\pm$ 1.3 | 5.05 $\pm$ 2.23 | 9.96 $\pm$ 6.71 |
| | DNAC | **1.00[0.99 - 1.00]** | 0.99[0.97 - 0.99] | 0.95[0.92 - 0.96] | **1.00[1.00 - 1.00]** |
| | | 4.52 $\pm$ 0.77 | 4.46 $\pm$ 1.0 | 5.05 $\pm$ 2.19 | 10.81 $\pm$ 4.63 |
| ExMSRNet | Linear | 0.81[0.76 - 0.84] | 0.99[0.92 - 1.00] | 0.91[0.90 - 0.93] | 0.99[0.94 - 1.00] |
| | | 5.01 $\pm$ 2.32 | 4.82 $\pm$ 1.52 | 4.28 $\pm$ 0.84 | 9.86 $\pm$ 6.41 |
| | DNAC | **1.00[0.99 - 1.00]** | **1.00[0.99 - 1.00]** | 0.93[0.92 - 0.94] | **1.00[1.00 - 1.00]** |
| | | 5.85 $\pm$ 1.07 | 5.18 $\pm$ 0.98 | 4.38 $\pm$ 0.57 | 10.33 $\pm$ 4.04 |
| Ground-Truth | | 1.00 | 1.00 | 0.99 | 1.00 |

## 6.2 Scarce-Data Regime

To evaluate data efficiency, we train each method using only a fraction of the synthetic training set while keeping the validation and test splits unchanged. In this low-data regime, MSRNet variants outperform NPU-based baselines and NALU, while remaining slightly below NMU in raw predictive accuracy.

These results indicate that structured feature selection and operator constraints continue to provide strong generalization under limited supervision, even though NMU retains a modest edge while consuming higher energy.

## 6.3 Hyperparameter Sensitivity and Hidden-Dimension Robustness

We further analyze sensitivity around the default ExMSRNet configuration. We vary each regularization hyperparameter separately (DL, order, sparsity) over a percentage scale of the training variance, while fixing the other two to 0, and observe stable trends across a broad range of hidden dimensions (see Figure 5). We conclude that while the addition of these losses result in an increase in performance, adding too much of any hyperparameter results in a drastic loss in performance.

## 6.4 Impact of varying Hidden dimensions

Table 2 shows the impact of varying the hidden dimension of MSRNet on the mean test $R^2$ of synthetic datasets. As the hidden dimension increases from 5 to 20, the mean test $R^2$ consistently degrades. A larger

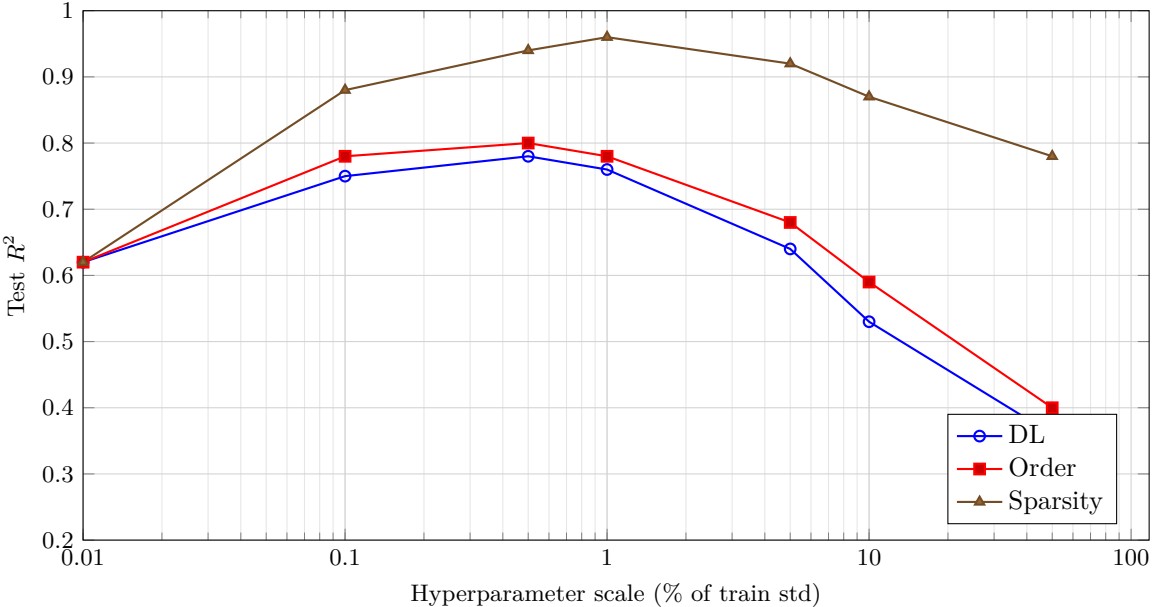

Figure 5: Hyperparameter sensitivity of MSRNet on Large Dimensions dataset. Each curve varies one hyperparameter (DL, order, sparsity) from 10% to 100% of training-variance scaling, while the other two are fixed to 0.

Table 2: Robustness to hidden dimension in MSRNet. Results are mean test $R^2$ on synthetic datasets.

| Hidden Dimension | Test $R^2$ | Expr. Sparsity ↓ | Energy ($10^{-3}$ Wh) ↓ | Entropy ↓ |
|---|---|---|---|---|
| 5 | 0.99 | 2.4 | 5.99 | 0.15 |
| 10 | 0.95 | 3.2 | 7.10 | 0.21 |
| 15 | 0.90 | 5.1 | 10.79 | 0.33 |
| 20 | 0.82 | 8.0 | 15.86 | 0.56 |

hidden dimension also leads to a loss in expression sparsity, which is the average number of inputs required to get the answer. Energy consumption follows a similar upward trajectory, being significantly lower for smaller hidden dimension. This could be attributed to two factors: Less number of trainable parameters, and early convergence. Furthermore, structural confidence, measured by entropy, decreases as the hidden dimension expands.

### 6.5 Taylor-Series Simulation of Exponential and Sine Functions

Even without explicit transcendental operator representational capability, these models simulate nonlinear functions through sparse power compositions that align with Taylor-series expansion of the respective operation. For example,

$$\exp(x) \approx 1 + x + \frac{x^2}{2!} + \frac{x^3}{3!} \qquad \sin(x) \approx x - \frac{x^3}{3!} \tag{27}$$

Empirically, all the tested models are capable of recovering compact polynomial-like approximations on **Exp** and **Sin** tasks. MSRNet exhibits this approximation on almost every run. ExMSRNet attains more compact symbolic forms through discrete `log` and `exp` gating that (de)activates log-domain processing and exponential reconstruction when beneficial.

On SRBench 2025, MSRNet variants achieves competitive performance while consuming significantly less energy than other symbolic regression methods. See Figures 6-8 for per method-dataset results and energy usage.

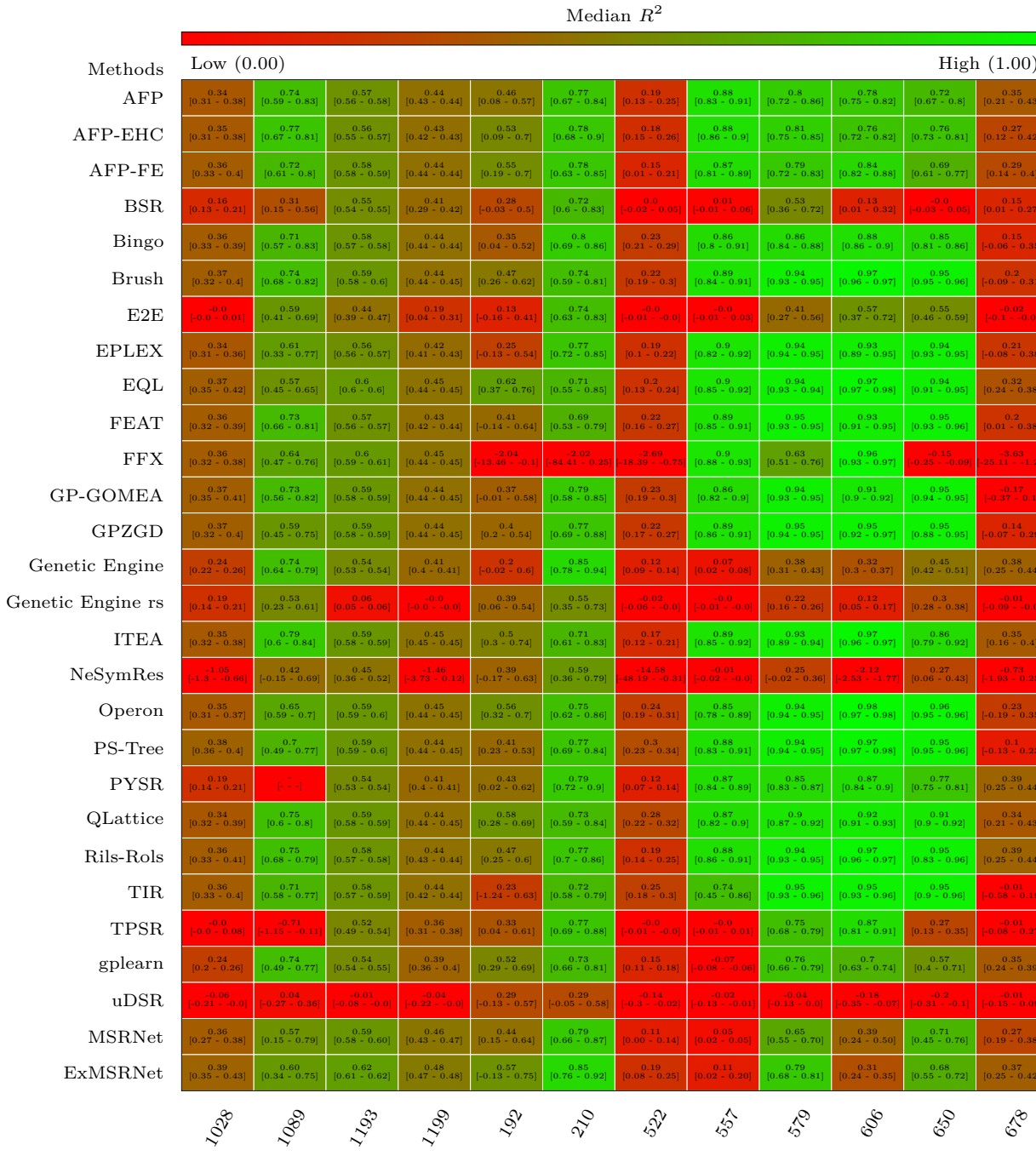

Figure 6: SRBench 2025 black-box track heatmap across various symbolic-regression baselines and MSRNet variants (one MSRNet unit used). Cell color represents median performance by a method in their respective datasets.

On AI Feynman I, II, and III, MSRNet variants recovers all target equations in our evaluation protocol. For these equations, we used a wider set of exponential candidates. We consider an equation to be "recovered" if the model weights result in an equivalent equation. For trigonometric and exponential relations, we

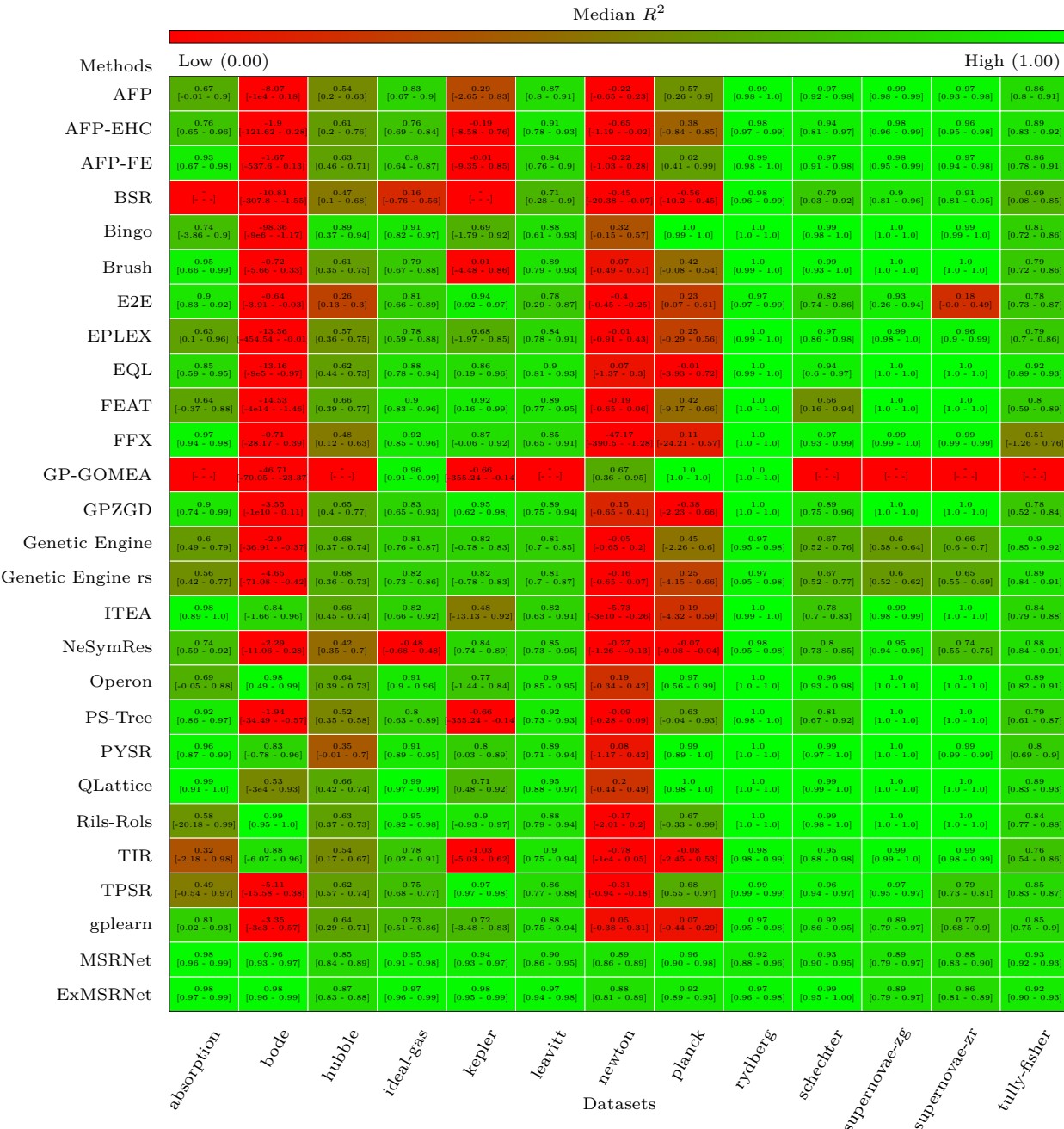

Figure 7: SRBench 2025 fundamental-equation track heatmap across various symbolic-regression baselines and MSRNet variants (varying number of MSR units used). Cell color represents median performance by a method in their respective datasets.

consider their equivalent taylor-approximation to be valid. Inclusion of scientific constants as extra input dimensions results in better recovery of equations. However, we have not included them in our experiments. While most equations are solved with a single or a double module, a small subset of equations requires three modules. This indicates that the architecture is expressive enough for complex symbolic forms while

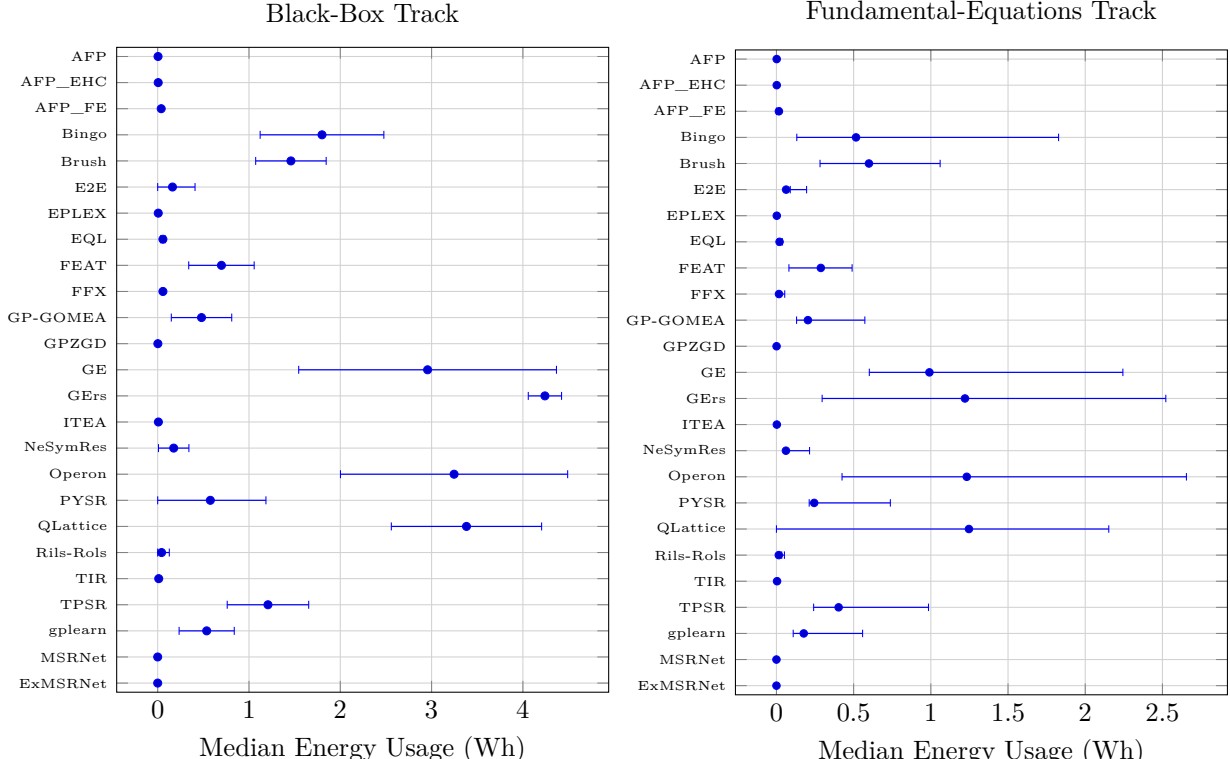

Figure 8: SRBench 2025 energy usage (kWh) comparison across various symbolic-regression baselines with MSRNet variants. Left: Black-box track. Right: Fundamental equations track.

remaining structurally compact. Detailed per-equation recovery logs (ground-truth equation and number of modules used) are reported in Appendix A.1.

## 6.6 Energy Efficiency

Across benchmarks, MSRNet requires substantially less energy than search-heavy symbolic regression alternatives and remains efficient relative to neural arithmetic baselines. Energy is measured using `eco2ai` on an NVIDIA Tesla P100 GPU for all compared neural methods (Budennyy et al., 2023; NVIDIA Corporation, 2016). This efficiency follows from two design choices: sparse feature gating, and end-to-end gradient training without expensive combinatorial search.

## 6.7 Result Summary

Our experiments suggest that MSRNet variants yield a strong balance of accuracy, symbolic fidelity, and computational efficiency across synthetic and standard symbolic-regression benchmarks, while consuming significantly less energy than other methods. This makes MSRNet a practical choice for scientific governing equation recovery on high-dimensional data. Further, under scarce-data training, MSRNet performs better than NALU and NPU baselines while performing equivalent or slightly worse than NMU. Additional analyses indicate stable behaviour across reasonable hyperparameter choices and hidden-dimension settings. For transcendental functions like sine and exponential functions, the MSRNet and other multiplicative models effectively learn sparse polynomial-like approximations corresponding to their Taylor-series expansions. Furthermore, MSRNet variants achieve competitive performance on the SRBench 2025 dataset, and 100% equation recovery with Feynman datasets, while requiring substantially less energy than traditional search-heavy symbolic regression alternatives.

## 7 Limitations

MSRNet is designed to prioritize interpretability through explicit structural constraints. As a consequence, it intentionally trades expressivity for structural confidence. Therefore, several limitations must be acknowledged.

First, the hypothesis space is restricted to arithmetic expressions composed of a predefined exponential vocabulary. Functions requiring dense interactions, highly nonlinear compositions, or non-rational exponents fall outside the representational scope of the model. While this restriction is central to interpretability, it limits applicability to domains where arithmetic structure is a reasonable prior.

Second, identifiability depends on sufficient data coverage. When multiple arithmetic expressions are functionally equivalent over the observed input domain, the model may converge to any of these representations. This limitation is inherent to equation discovery and symbolic regression methods and cannot be resolved purely through architectural constraints.

Third, the discrete operator sets used in this work are fixed a priori. Although they are chosen to cover a broad range of common arithmetic operations, extending or adapting the operator vocabulary may be necessary for certain application domains. Learning the operator set itself remains an open problem.

Finally, while description-length and entropy regularization improve stability and structural recovery in practice, they do not provide formal guarantees of optimal symbolic recovery. Theoretical guarantees remain conditional on assumptions regarding noise, data distribution, and functional sparsity.

These limitations reflect deliberate design choices rather than implementation shortcomings, and they define the regime in which the proposed approach is most effective.

## 8 Conclusion

We demonstrates that relying solely on arithmetic inductive biases is insufficient to guarantee the extraction of sparse, interpretable symbolic equations from neural networks. Instead, structural constraints should be explicitly embedded into the architecture. To this end, we introduced Multiplicative Symbolic Regression Networks (MSRNet) and its extended variant, ExMSRNet. These architectures utilize a discrete Neural Accumulator (DNAC) for sparse additive feature selection, coupled with a discrete RealNPU core to model multiplicative and exponential interactions. By constraining the optimization space to discrete operator selections and using description-length regularization, MSRNet effectively mitigates unstable optimization behaviors and gradient starvation. Empirical results suggest that MSRNet variants achieves a balance between predictive accuracy, symbolic fidelity, and computational efficiency. These models not only scale more favorably compared to explicit candidate-library methods but also yield structurally compact formulas that do not require post-hoc surrogate approximations.

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

## A  Appendix

### A.1  AI Feynman I/II/III: Per-Equation Recovery Logs

Tables 3–5 are structured for full per-equation reporting. For each equation ID, we record: (i) ground-truth symbolic form, (ii) number of modules/layers used by MSRNet variants, and (iii) whether the equation is recovered in more than 50% of the cases across 30 runs.

Table 3: AI Feynman I: per-equation recovery log.

| Set | Equation ID | Ground-Truth Equation | MSRNet | | ExMSRNet | |
|---|---|---|---|---|---|---|
| | | | Recovered | Layers | Recovered | Layers |
| I | 6-2 | `exp(-(theta/sigma)**2/2)/(sqrt(2*pi)*sigma)` | ✓ | 2 | ✓ | 2 |
| I | 6-2a | `exp(-theta**2/2)/sqrt(2*pi)` | ✓ | 2 | ✓ | 2 |
| I | 6-2b | `exp(-((theta-theta1)/sigma)**2/2)/(sqrt(2*pi)*sigma)` | ✓ | 2 | ✓ | 2 |
| I | 8-14 | `sqrt((x2-x1)**2+(y2-y1)**2)` | ✓ | 2 | ✓ | 2 |
| I | 9-18 | `G*m1*m2/((x2-x1)**2+(y2-y1)**2+(z2-z1)**2)` | ✓ | 2 | ✓ | 2 |
| I | 10-7 | `m_0/sqrt(1-v**2/c**2)` | ✓ | 2 | ✓ | 2 |
| I | 11-19 | `x1*y1+x2*y2+x3*y3` | ✓ | 1 | ✓ | 1 |
| I | 12-1 | `mu*Nn` | ✓ | 1 | ✓ | 1 |
| I | 12-2 | `q1*q2*r/(4*pi*epsilon*r**3)` | ✓ | 1 | ✓ | 1 |
| I | 12-4 | `q1*r/(4*pi*epsilon*r**3)` | ✓ | 1 | ✓ | 1 |
| I | 12-5 | `q2*Ef` | ✓ | 1 | ✓ | 1 |
| I | 12-11 | `q*(Ef+B*v*sin(theta))` | ✓ | 1 | ✓ | 1 |
| I | 13-4 | `1/2*m*(v**2+u**2+w**2)` | ✓ | 1 | ✓ | 1 |
| I | 13-12 | `G*m1*m2*(1/r2-1/r1)` | ✓ | 1 | ✓ | 1 |
| I | 14-3 | `m*g*z` | ✓ | 1 | ✓ | 1 |
| I | 14-4 | `1/2*k_spring*x**2` | ✓ | 1 | ✓ | 1 |
| I | 15-3t | `(t-u*x/c**2)/sqrt(1-u**2/c**2)` | ✓ | 2 | ✓ | 2 |
| I | 15-3x | `(x-u*t)/sqrt(1-u**2/c**2)` | ✓ | 2 | ✓ | 2 |
| I | 15-10 | `m_0*v/sqrt(1-v**2/c**2)` | ✓ | 2 | ✓ | 2 |
| I | 16-6 | `(u+v)/(1+u*v/c**2)` | ✓ | 2 | ✓ | 2 |
| I | 18-4 | `(m1*r1+m2*r2)/(m1+m2)` | ✓ | 1 | ✓ | 1 |

| Set | Equation ID | Ground-Truth Equation | MSRNet | | ExMSRNet | |
|---|---|---|---|---|---|---|
| | | | Recovered | Layers | Recovered | Layers |
| I | 18-12 | `r*F*sin(theta)` | ✓ | 1 | ✓ | 1 |
| I | 18-14 | `m*r*v*sin(theta)` | ✓ | 1 | ✓ | 1 |
| I | 24-6 | `1/2*m*(omega**2+omega_0**2)*1/2*x**2` | ✓ | 1 | ✓ | 1 |
| I | 25-13 | `q/C` | ✓ | 1 | ✓ | 1 |
| I | 26-2 | `arcsin(n*sin(theta2))` | ✓ | 2 | ✓ | 2 |
| I | 27-6 | `1/(1/d1+n/d2)` | ✓ | 1 | ✓ | 1 |
| I | 29-4 | `omega/c` | ✓ | 1 | ✓ | 1 |
| I | 29-16 | `sqrt(x1**2+x2**2-2*x1*x2*cos(theta1-theta2))` | ✓ | 2 | ✓ | 2 |
| I | 30-3 | `Int_0*sin(n*theta/2)**2/sin(theta/2)**2` | ✓ | 2 | ✓ | 2 |
| I | 30-5 | `arcsin(lambd/(n*d))` | ✓ | 2 | ✓ | 2 |
| I | 32-5 | `q**2*a**2/(6*pi*epsilon*c**3)` | ✓ | 1 | ✓ | 1 |
| I | 32-17 | `(1/2*epsilon*c*Ef**2)*(8*pi*r**2/3)*(omega**4/(omega**2-omega_0**2)**2)` | ✓ | 2 | ✓ | 2 |
| I | 34-1 | `omega_0/(1-v/c)` | ✓ | 1 | ✓ | 1 |
| I | 34-8 | `q*v*B/p` | ✓ | 1 | ✓ | 1 |
| I | 34-14 | `(1+v/c)/sqrt(1-v**2/c**2)*omega_0` | ✓ | 2 | ✓ | 2 |
| I | 34-27 | `(h/(2*pi))*omega` | ✓ | 1 | ✓ | 1 |
| I | 37-4 | `I1+I2+2*sqrt(I1*I2)*cos(delta)` | ✓ | 1 | ✓ | 1 |
| I | 38-12 | `4*pi*epsilon*(h/(2*pi))**2/(m*q**2)` | ✓ | 1 | ✓ | 1 |
| I | 39-1 | `3/2*pr*V` | ✓ | 1 | ✓ | 1 |
| I | 39-11 | `1/(gamma-1)*pr*V` | ✓ | 2 | ✓ | 2 |
| I | 39-22 | `n*kb*T/V` | ✓ | 1 | ✓ | 1 |
| I | 40-1 | `n_0*exp(-m*g*x/(kb*T))` | ✓ | 2 | ✓ | 2 |
| I | 41-16 | `h/(2*pi)*omega**3/(pi**2*c**2*(exp((h/(2*pi))*omega/(kb*T))-1))` | ✓ | 2 | ✓ | 2 |
| I | 43-16 | `mu_drift*q*Volt/d` | ✓ | 1 | ✓ | 1 |
| I | 43-31 | `mob*kb*T` | ✓ | 1 | ✓ | 1 |
| I | 43-43 | `1/(gamma-1)*kb*v/A` | ✓ | 2 | ✓ | 2 |
| I | 44-4 | `n*kb*T*ln(V2/V1)` | ✓ | 3 | ✓ | 2 |
| I | 47-23 | `sqrt(gamma*pr/rho)` | ✓ | 1 | ✓ | 1 |
| I | 48-2 | `m*c**2/sqrt(1-v**2/c**2)` | ✓ | 2 | ✓ | 2 |
| I | 50-26 | `x1*(cos(omega*t)+alpha*cos(omega*t)**2)` | ✓ | 3 | ✓ | 3 |

Table 4: AI Feynman II: per-equation recovery log.

| Set | Equation ID | Ground-Truth Equation | MSRNet | | ExMSRNet | |
|---|---|---|---|---|---|---|
| | | | Recovered | Layers | Recovered | Layers |
| II | 2-42 | `kappa*(T2-T1)*A/d` | ✓ | 1 | ✓ | 1 |
| II | 3-24 | `Pwr/(4*pi*r**2)` | ✓ | 1 | ✓ | 1 |
| II | 4-23 | `q/(4*pi*epsilon*r)` | ✓ | 1 | ✓ | 1 |
| II | 6-11 | `1/(4*pi*epsilon)*p_d*cos(theta)/r**2` | ✓ | 1 | ✓ | 1 |
| II | 6-15a | `p_d/(4*pi*epsilon)*3*z/r**5*sqrt(x**2+y**2)` | ✓ | 2 | ✓ | 2 |
| II | 6-15b | `p_d/(4*pi*epsilon)*3*cos(theta)*sin(theta)/r**3` | ✓ | 1 | ✓ | 1 |
| II | 8-7 | `3/5*q**2/(4*pi*epsilon*d)` | ✓ | 1 | ✓ | 1 |
| II | 8-31 | `epsilon*Ef**2/2` | ✓ | 1 | ✓ | 1 |

| Set | Equation ID | Ground-Truth Equation | MSRNet | | ExMSRNet | |
|-----|-------------|----------------------|--------|--------|----------|--------|
| | | | Recovered | Layers | Recovered | Layers |
| II | 10-9 | sigma_den/epsilon*1/(1+chi) | ✓ | 1 | ✓ | 1 |
| II | 11-3 | q*Ef/(m*(omega_0**2-omega**2)) | ✓ | 1 | ✓ | 1 |
| II | 11-20 | n_rho*p_d**2*Ef/(3*kb*T) | ✓ | 1 | ✓ | 1 |
| II | 11-27 | n*alpha/(1-(n*alpha/3))*epsilon*Ef | ✓ | 2 | ✓ | 2 |
| II | 11-28 | 1+n*alpha/(1-(n*alpha/3)) | ✓ | 2 | ✓ | 2 |
| II | 13-17 | 1/(4*pi*epsilon*c**2)*2*I/r | ✓ | 1 | ✓ | 1 |
| II | 13-23 | rho_c_0/sqrt(1-v**2/c**2) | ✓ | 1 | ✓ | 1 |
| II | 13-34 | rho_c_0*v/sqrt(1-v**2/c**2) | ✓ | 1 | ✓ | 1 |
| II | 15-4 | -mom*B*cos(theta) | ✓ | 1 | ✓ | 1 |
| II | 15-5 | -p_d*Ef*cos(theta) | ✓ | 1 | ✓ | 1 |
| II | 21-32 | q/(4*pi*epsilon*r*(1-v/c)) | ✓ | 1 | ✓ | 1 |
| II | 24-17 | sqrt(omega**2/c**2-pi**2/d**2) | ✓ | 2 | ✓ | 2 |
| II | 27-16 | epsilon*c*Ef**2 | ✓ | 1 | ✓ | 1 |
| II | 27-18 | epsilon*Ef**2 | ✓ | 1 | ✓ | 1 |
| II | 34-2 | q*v*r/2 | ✓ | 1 | ✓ | 1 |
| II | 34-2a | q*v/(2*pi*r) | ✓ | 1 | ✓ | 1 |
| II | 34-11 | g_*q*B/(2*m) | ✓ | 1 | ✓ | 1 |
| II | 34-29a | q*h/(4*pi*m) | ✓ | 1 | ✓ | 1 |
| II | 34-29b | g_*mom*B*Jz/(h/(2*pi)) | ✓ | 1 | ✓ | 1 |
| II | 35-18 | n_0/(exp(mom*B/(kb*T))+exp(-mom*B/(kb*T))) | ✓ | 3 | ✓ | 3 |
| II | 35-21 | n_rho*mom*tanh(mom*B/(kb*T)) | ✓ | 2 | ✓ | 2 |
| II | 36-38 | mom*H/(kb*T)+(mom*alpha)/(epsilon*c**2*kb*T)*M | ✓ | 1 | ✓ | 1 |
| II | 37-1 | mom*(1+chi)*B | ✓ | 1 | ✓ | 1 |
| II | 38-3 | Y*A*x/d | ✓ | 1 | ✓ | 1 |
| II | 38-14 | Y/(2*(1+sigma)) | ✓ | 2 | ✓ | 2 |

Table 5: AI Feynman III: per-equation recovery log.

| Set | Equation ID | Ground-Truth Equation | MSRNet | | ExMSRNet | |
|-----|-------------|----------------------|--------|--------|----------|--------|
| | | | Recovered | Layers | Recovered | Layers |
| III | 4-32 | 1/(exp((h/(2*pi))*omega/(kb*T))-1) | ✓ | 3 | ✓ | 3 |
| III | 4-33 | (h/(2*pi))*omega/(exp((h/(2*pi))*omega/(kb*T))-1) | ✓ | 3 | ✓ | 3 |
| III | 7-38 | 2*mom*B/(h/(2*pi)) | ✓ | 1 | ✓ | 1 |
| III | 8-54 | sin(E_n*t/(h/(2*pi)))**2 | ✓ | 3 | ✓ | 3 |
| III | 9-52 | (p_d*Ef*t/(h/(2*pi)))*sin((omega-omega_0)*t/2)**2/((omega-omega_0)*t/2)**2 | ✓ | 3 | ✓ | 3 |
| III | 10-19 | mom*sqrt(Bx**2+By**2+Bz**2) | ✓ | 2 | ✓ | 2 |
| III | 12-43 | n*(h/(2*pi)) | ✓ | 1 | ✓ | 1 |
| III | 13-18 | 2*E_n*d**2*k/(h/(2*pi)) | ✓ | 1 | ✓ | 1 |
| III | 14-14 | I_0*(exp(q*Volt/(kb*T))-1) | ✓ | 1 | ✓ | 1 |
| III | 15-12 | 2*U*(1-cos(k*d)) | ✓ | 1 | ✓ | 1 |
| III | 15-14 | (h/(2*pi))**2/(2*E_n*d**2) | ✓ | 1 | ✓ | 1 |
| III | 15-27 | 2*pi*alpha/(n*d) | ✓ | 1 | ✓ | 1 |
| III | 17-37 | beta*(1+alpha*cos(theta)) | ✓ | 1 | ✓ | 1 |
| III | 19-51 | -m*q**4/(2*(4*pi*epsilon)**2*(h/(2*pi))**2)*(1/n**2) | ✓ | 1 | ✓ | 1 |
| III | 21-20 | -rho_c_0*q*A_vec/m | ✓ | 1 | ✓ | 1 |

