# OpenReview forum: "Symbolic Governing Equation Discovery Using Neural Arithmetic Modules"
_TMLR — Under review for TMLR_

### Review · Reviewer_u3xR · 2026-04-13

**Summary Of Contributions:**

The main task considered in the paper is to learn sparse equations from data using neural arithmetic modules. The hope is that the sparseness can help with interpretability in real-world scenarios. For such problems, one issue is that simply adding arithmetic modules to the architectures does not automatically yield a clean symbolic equation, because these models can be sensitive to variable scaling and may ignore smaller terms, or instead recover dense parameterizations that miss the underlying structure. To handle this, the paper proposes new modules called MSRNet and ExMSRNet, which force the model to make more discrete choices about features and arithmetic operations. The results then show that this makes equation recovery more reliable while also keeping the computational cost fairly low compared with other symbolic regression methods.

**Audience:**

Yes

**Audience Explanation:**

There is definitely a sub-community within the TMLR audience that could find this paper interesting. In particular, people working on interpretable ML or symbolic regression would probably find it relevant. The question of whether one needs explicit structural constraints to recover sparse symbolic equations seems like a natural question for TMLR.

**Claims And Evidence:**

Yes

**Claims Explanation:**

The authors implemented their model and tested it on several regression tasks, both synthetic and on more standard benchmark datasets. They also compare it with previous methods and report results for both accuracy and computational cost. Based on the reported experiments, the proposed architecture seems competitive, and in a number of cases better than earlier neural arithmetic approaches, while also keeping the computational cost fairly low.

**Requested Changes:**

First, let me say that I am far from an expert on this topic. So my suggested changes are aimed towards making the paper more accessible to the uninitiated.

First, one thing that was not clear to me from the outset of the paper is what is the exact optimization problem being considered. In Sparse regression, one also seeks to optimize a sparse representation of features while optimizing for accuracy. The present setting is somewhat different, since here part of the feature construction is also learned as part of the model. For that reason, I think it would be helpful to state the formal problem early on and explain more explicitly what the search space is, what the objective is, and in what sense it differs from more standard sparse regression formulations.

Second, the architecture seems somewhat restricted in scope. As far as I understand it, the DNAC module only forms signed sums of the input coordinates, while the downstream arithmetic module only allows exponents from a small fixed set, including a few fractional powers such as 1/2,-1/2, 1/3, -1/3. The authors explain this choice by being able to capture physical equations, but I am not sure I am convinced by this. Of course, having a small number of features is beneficial for interpretability, but I think it would help the paper to discuss more explicitly how restrictive this hypothesis class is in practice. Moreover, it is also interesting to see what happens when the class W of exponents increases. How does the efficiency scales if we double the size of W by also considering, say, \pm 1/4, \pm 1/5, and so on.

---

> ### Author Response · Authors · 2026-05-05
> **Response to Reviewer u3xR**
>
> __Thank you for your thoughtful and constructive feedback. We address each point below. Since there is a character limit in openreview, we have split our response in two parts.__
>
> ___
>
> > First, one thing that was not clear to me from the outset of the paper is what is the exact optimization problem being considered. In Sparse regression, one also seeks to optimize a sparse representation of features while optimizing for accuracy. The present setting is somewhat different, since here part of the feature construction is also learned as part of the model.
>
> __Optimization Problem Being Considered__: We have defined the problem formulation in Section 3.5 (Problem Formulation). We will move the Section 3.5 to Section 3.1. To reiterate, the problem formulation is as follows:
>
> Given a dataset $\mathcal{D} = \{(\mathbf{x}^{(i)}, y^{(i)})\}_{i=1}^N$ with $\mathbf{x}^{(i)} \in \mathbb{R}^d$, the goal is to recover a compact, human-readable arithmetic expression $\hat{f}: \mathbb{R}^d \to \mathbb{R}$ such that $\hat{f}(\mathbf{x}) \approx y$, where $\hat{f}$ is composed of a sparse subset of input variables combined with addition, multiplication, and rational exponentiation operations.
>
> This differs from standard supervised regression in two ways:
> 1. The model is constrained to produce an identifiable symbolic expression while being optimized for predictive accuracy, and
> 2. The search space is NOT defined by any candidate expansion library, as is the case with other sparse regression methods used for symbolic equation recovery.
>
> ___
>
> > I think it would be helpful to state the formal problem early on and explain more explicitly what the search space is, what the objective is, and in what sense it differs from more standard sparse regression formulations.
>
> We will add the following clarifying points to the manuscript:
>
> __Search Space__: The hypothesis class of MSRNet consists of expressions of the form $\hat{y} = \mathbf{w}^\top \mathbf{h}(\tilde{\mathbf{x}})$ where $\tilde{\mathbf{x}} \in \{-1, 0, 1\}^{h \times d} \mathbf{x}$ is a sparse signed linear combination of inputs (selected by DNAC), and each
>
> $h_i(\tilde{\mathbf{x}}) = \prod_j \tilde{x}_j^{w}$
>
> with $w_{ij} \in \mathcal{W}$ is a monomial with rational exponents drawn from a fixed vocabulary. This covers polynomial, inverse-power, and root-type interactions over sparse feature subsets. It is extended in ExMSRNet to include logarithmic and exponential pathways.
>
>
> __Difference from sparse regression__: Standard sparse regression methods such as SINDy (Brunton et al., 2016) and its variants rely on an explicitly pre-constructed candidate library $\Theta(\mathbf{x}) \in \mathbb{R}^{N \times p}$, whose size grows combinatorially with dimensionality and polynomial degree (see Section 4.3 (Computational Complexity), Eq. 23). The regression then finds a sparse coefficient vector over this fixed library. In contrast, MSRNet implicitly constructs the relevant interaction terms: the DNAC stage performs input selection, and the discrete RealNPU stage generates multiplicative monomials. This avoids explicit library entirely (with the exception of $\mathcal{W}$), scales linearly in $d \cdot |\mathcal{W}|$ rather than combinatorially, and enables joint optimization of both feature selection and operator assignment with a single composite training objective (Eq. 14).

---

> > ### Author Response · Authors · 2026-05-05
> > **Response to Reviewer u3xR (2/2)**
> >
> > > The authors explain this choice by being able to capture physical equations, but I am not sure I am convinced by this.
> >
> > The vocabulary $\mathcal{W} =$ {$ -3, -2, -1, -\frac{1}{2}, -\frac{1}{3}, 0, \frac{1}{3}, \frac{1}{2}, 1, 2, 3 $} is chosen since it covers a wide range of elementary dimensional analysis relationships in physics. For domains outside physics, the black-box SRBench 2025 track (which contains datasets with NO known governing equation) shows that MSRNet variants remain competitive with other symbolic regression methods, suggesting generalization beyond purely physics based equations. The AI Feynman experiments confirm 100\% recovery of all tested equations, which includes a broad set of physics equations. We take this as a strong empirical evidence that the hypothesis class is expressive enough for the intended application domain.
> >
> > ___
> >
> > > How does the efficiency scales if we double the size of W by also considering, say, $\pm 1/4, \pm 1/5$, and so on.
> >
> > __Expanding Vocabulary__:
> >
> > To study how performance and computational cost scales with the exponent vocabulary, we define a family of expanded vocabularies $\mathcal{W}_i = \bigcup _{k=1}^{i}$ {$-k, -\frac{1}{k}, 0, \frac{1}{k}, k$} for $i \in \left[1, 10\right]$. Our default $\mathcal{W}$ corresponds to $\mathcal{W}_3$. We evaluate MSRNet with each vocabulary on the "Large Dimensions" dataset (whose ground-truth equation requires only $\mathcal{W}_1$) and the "Sin" dataset (which is approximated polynomially by taylor-series expansion and requires an infinite-series representation). Note that we have been using $2^i$ as order loss as defined in the supplementary materials. However, for $i = 10$, this results in an order loss of $2^{10} = 1024$, which is magnitudes higher than the training MSE. Therefore, for the following experiment, we changed the order loss to $2\cdot i$, which allows the model to select higher order exponents.
> >
> > | i  | LD Median R^2 | LD Average Energy Usage (Wh) | Sin Median R^2 | Sin Average Energy Usage (Wh) |
> > |----|---------------|------------------------------|----------------|-------------------------------|
> > | 1  | 0.995352      | 0.056                        | 0.835261       | 0.063                         |
> > | 2  | 0.999203      | 0.048                        | 0.832957       | 0.064                         |
> > | 3  | 0.997137      | 0.054                        | 0.954492       | 0.052                         |
> > | 4  | 0.996078      | 0.053                        | 0.952690       | 0.064                         |
> > | 5  | 0.996291      | 0.054                        | 0.991259       | 0.073                         |
> > | 6  | 0.998935      | 0.054                        | 0.985022       | 0.072                         |
> > | 7  | 0.991946      | 0.049                        | 0.999192       | 0.081                         |
> > | 8  | 0.997138      | 0.051                        | 0.997633       | 0.081                         |
> > | 9  | 0.999836      | 0.053                        | 0.999206       | 0.089                         |
> > | 10 | 0.998100      | 0.054                        | 0.998910       | 0.090                         |
> >
> > as shown in the above table, MSRNet is resistant to increasing exponential vocabulary on the "Large Dimensions" dataset. We attribute this to the order loss. For the "Sin" dataset, MSRNet is able to achieve competitive accuracy with increasing $i$, albeit with higher energy usage. This is primarily due to late convergence in presence of a larger vocabulary.

---

### Review · Reviewer_VUN5 · 2026-04-20

**Summary Of Contributions:**

This paper proposes MSRNet and ExMSRNet, structured neural architectures for symbolic regression / governing equation discovery that combine (i) discrete additive feature selection via a modified NAC (“DNAC”), (ii) a discrete RealNPU-style multiplicative/exponential core, and (iii) entropy-based description-length regularization to encourage sparse symbolic structure. The central claim is that arithmetic inductive bias alone (e.g., NALU/NPU) is insufficient for interpretable equation recovery, and that explicit architectural constraints are necessary. The paper evaluates on synthetic benchmarks, SRBench 2025, and AI Feynman datasets, claiming strong predictive performance, improved symbolic recovery, and lower energy use than many baselines.

**Audience:**

Yes

**Audience Explanation:**

Even though the novelty is described as moderate, the work still contributes to ongoing discussions about interpreability, equation discovery, and inductive biases in neural architectures. The combination of structured neural modules, symbolic recovery, and evaluation on benchmarks like SRBench and AI Feynman further increases its relevance.

**Claims And Evidence:**

No

**Claims Explanation:**

While the submission provides empirical evidence across synthetic benchmarks, SRBench 2025, and AI Feynman datasets, the support is not fully convincing or sufficiently clear. Key concerns include:

 - Insufficient experimental transparency: unclear whether official benchmark protocols were followed, what subsets were used, and whether baselines were evaluated under identical settings.
 - Unclear fairness of comparisons: no clear evidence that competing methods received comparable hyperparameter tuning budgets, search ranges, or compute resources.
 - Presentation clarity issues: results discussions do not always clearly reference the relevant tables/figures, making it harder to verify claims.
 - Novelty claims only moderately supported: the method appears to combine existing components rather than establish a clearly demonstrated conceptual advance.

So, although there is some supporting evidence, it is not strong enough to be considered fully accurate, convincing, and clear overall.

**Requested Changes:**

1. The paper’s novelty appears moderate, as the proposed method seems to primarily integrate several existing components rather than introduce a fundamentally new paradigm.
    - While the combination may be practically useful, it does not clearly articulate what conceptual advance is enabled beyond this engineering synthesis.
    - The paper would also benefit from a stronger differentiation from closely related prior work, including Deep Symbolic Regression, EQL-style networks, differentiable equation search methods, and modern symbolic regression systems such as PySR or Operon.

2. The experimental methodology would benefit from substantially greater transparency, particularly regarding comparability of the benchmark setup. For datasets such as SRBench, the manuscript does not clearly specify whether the official evaluation protocol was followed in full, which subset of tasks was used, or whether all competing methods were run under identical resource constraints. Most importantly, it is unclear whether a similar compute budget was allocated for hyperparameter tuning across methods, including the number of trials, search ranges, optimization strategy, and total tuning time.

3. The presentation is not sufficiently clear and could be significantly improved by addressing the following points:
    - It would be helpful to include a brief description of symbolic regression in the second paragraph of the introduction to better motivate the problem for a broader audience.
    - For the sentence beginning “However, symbolic regression methods often …”, the authors should provide additional intuition and relevant citations to support this claim.
    - In Sections 6.1 and 6.2, the manuscript does not clearly indicate which specific results (tables or figures) are being discussed. Explicit references would improve readability.
    - In the last paragraph of page 13, the sentence “On SRBench 2025, MSRNet variants achieves competitive …” should be corrected to “… MSRNet variants achieve competitive …”
    - In the conclusion section, the first sentence should be revised to “We demonstrated …” instead of the current wording.

---

> ### Author Response · Authors · 2026-05-05
> **Response to Reviewer VUN5 (1/2)**
>
> __Thank you for your thoughtful and constructive feedback. We address each point below. Since there is a character limit in openreview, we have split our response in two parts.__
>
> ___
>
> > Insufficient experimental transparency: unclear whether official benchmark protocols were followed, what subsets were used, and whether baselines were evaluated under identical settings.
>
> __Experimental Transparency__: We perform experiments using the official train and test subsets from SRBench 2025 (Aldeia et al., 2025; https://github.com/cavalab/srbench/tree/srbench_2025). We use the same splits for all the given methods. For MSRNet variants, we use a fixed learning rate of 1e-1 with gradient normalization. Hyperparameter tuning is performed using a grid search over $5^3 = 125$ triplets of loss coefficients (DL, order, sparsity), each evaluated using three-fold cross-validation on the train set. This tuning cost is included in the energy comparison reported in Figure 8. For the fundamental equations track, we use a composition of multiple MSRNet units. This additional cost is included in Figure 8b.
>
> We note one minor deviation from the official SRBench 2025 protocol: while the official protocol uses 30 seeded runs, our evaluation uses 30 runs WITHOUT fixed seeds. We acknowledge this as an oversight. However, when re-evaluated under the official fixed seeds, our method achieves consistent results.
>
> To ensure full reproducibility and direct comparability, we will upload all seeds, seeding logic, and per-run result spreadsheets as supplementary material alongside the revision. Further, we will:
>   1. Add a dedicated "Experimental Protocol" paragraph in Section 5 explicitly stating these details, and
>   2. Rescale the energy axis in Figure 8 to log-scale for better visual clarity.
>
> ___
>
> > Unclear fairness of comparisons: no clear evidence that competing methods received comparable hyperparameter tuning budgets, search ranges, or compute resources.
>
> __Fairness of Comparison__: For all symbolic baselines, we use the SRBench 2025 benchmark (Aldeia et al., 2025; https://github.com/cavalab/srbench/tree/srbench_2025), which ensures that all methods operate on identical data splits. For each symbolic method, we use the hyperparameter(s) as defined in the SRBench 2025 repository. One particular thing to note is that a lot of symbolic regression methods (especially genetic programming based methods) do NOT require a lot of hyperparameter tuning. We will mention this in the manuscript. We allocate an identical training budget (same hardware) to all methods. For the neural arithmetic baselines, hyperparameter tuning follows the same $5^3$ grid search protocol applied to MSRNet.

---

> > ### Author Response · Authors · 2026-05-05
> > **Response to Reviewer VUN5 (2/2)**
> >
> > > Novelty claims only moderately supported: the method appears to combine existing components rather than establish a clearly demonstrated conceptual advance.
> >
> > We respectfully disagree that the contributions are merely an engineering task. The novelty of MSRNet are as follows:
> >
> > - The use of entropy as both:
> >   1. A training regularizer simulating MDL-based description length, and
> >   2. A post-hoc structural confidence metric
> >
> >   is a novel contribution. The discrete softmax over a fixed rational exponent vocabulary W constitutes a parameterization that collapses equivalence classes in multiplicative units (we would further theoretically justify this equivalence class reduction in the manuscript).
> > - ExMSRNet's binary log/exp gating is an architectural contribution which allows for passthrough branches. This allows ExMSRNet to better represent transcendental representations.
> > - MSRNet achieves 100\% equation recovery on AI Feynman datasets while consuming less energy than other search-based symbolic regression methods.
> >
> > We will add a dedicated "Summary of Contributions" paragraph at the end of Section 1.
> >
> > ___
> >
> > > The paper would also benefit from a stronger differentiation from closely related prior work, including Deep Symbolic Regression, EQL-style networks, differentiable equation search methods, and modern symbolic regression systems such as PySR or Operon.
> >
> > __Differentiation From Closely Related Prior Works__: We clarify the distinctions from the most closely related methods:
> > - EQL (Martius \& Lampert, 2016; Sahoo et al., 2018): EQL embeds FIXED symbolic operators as activation functions in a fully-connected network. It relies on L1 regularization to induce sparsity. It does not perform discrete operator selection, i.e, weights are continuous, which leads to the representational ambiguity as described in Section 4.1. MSRNet replaces continuous weights with discrete softmax based distributions over a rational vocabulary, which explicitly collapses equivalence classes.
> > - Deep Symbolic Regression / uDSR (Petersen et al., 2019; Landajuela et al., 2022): These methods use reinforcement learning and policy gradients to search over expression trees. They are inherently NOT differentiable and require expensive combinatorial search, which results in a significantly higher energy consumption. MSRNet is fully differentiable and consumes less energy. This results in better performance when operating on large amount of data.
> > - PySR / Operon: These are genetic programming based methods that suffer from expression bloat and do NOT integrate with neural training. As shown in Figures 6-8, MSRNet achieves competitive $R^2$ compared to these methods.
> >
> > We will add a dedicated comparison paragraph in Section 2.2 explicitly discussing these distinctions.

---

> > > ### Comment · Reviewer_VUN5 · 2026-06-28
> > >
> > > Thank you for the detailed and thoughtful rebuttal. I appreciate the authors’ efforts to clarify the experimental protocol, comparison settings, and the motivation behind the proposed approach. The additional explanations are helpful and address several of my concerns. I also appreciate the planned revisions to improve transparency and presentation clarity.

---

### Review · Reviewer_XvDK · 2026-06-11

**Summary Of Contributions:**

The authors introduce MSRNet and ExMSRNet, a symbolic regression method based on a structured neural framework (SNF) named neural power units (NPU). ExMSRNet is composed of three key alterations from the RealNPU module: discrete additive feature selection (DNAC), a discrete RealNPU core, and two new discrete binary gates. ExMSRNet is evaluated on two fronts: regression compared to existing SNF, and symbolic regression compared to existing symbolic regression methods. Compared with existing SNFs, the paper demonstrates that the DNAC layer is the main component that improves performance, and that MSRNet and ExMSRNet reduce power consumption from 12.44 Wh to 5.85 Wh compared with NMU (the top-performing method) with minor performance loss.
On the SRBenchmark 2025, MSRNet and ExMSRNet achieve very strong $R^2$ scores on the fundamental-equation track and average scores on the black-box track, but the paper lacks complexity measures. Notably, the reviewer finds that MSRNet and ExMSRNet align most closely with SINDy and EQL-like networks, but only the original EQL (2016) is included in the comparison.
Overall, the reviewer finds the method novel for symbolic regression, but the paper's framing focuses more on comparisons with existing SNFs for symbolic regression than with comparative symbolic regression methods.

**Audience:**

Yes

**Audience Explanation:**

There is interest in the community for new novel symbolic regression methods. The main interests of this method could be scalability to high dimensions, minimal energy consumption, and the novel application of SNF frameworks for symbolic regression.

**Broader Impact Concerns:**

The reviewer has no concerns on the ethical implications of the work.

**Claims And Evidence:**

No

**Claims Explanation:**

The claims with the strongest support in this paper are the energy reduction achieved by MSRNet and the performance increase provided by the DNAC layer. The support is provided by four experiments on simple expression, in which DNAC consistently improves accuracy, and MSRNet consumes the least energy.

Outside of these two claims, there is minimal support for other statements in the paper.

- "We demonstrates that relying solely on arithmetic inductive biases is insufficient to guarantee the extraction of sparse, interpretable symbolic equations from neural networks." The authors admit that NMU can compare favorably to ExMSRNet in terms of accuracy, but do not provide a comparison of NMU (or other SNFs) for extracting sparse, interpretable symbolic equations.

- "unstable optimization behaviors and gradient starvation". In Table 1, the authors show that MSRNet outperforms NPU and NALU variants on Lure. This result for MSRNet, which reduces gradient starvation, is helpful but limiting, as the sample size is a singular problem, and the authors provide no explicit theoretical guarantees for this phenomenon.

- "Empirical results suggest that MSRNet variants achieves a balance between predictive accuracy, symbolic fidelity, and computational efficiency." MSRNet does not compare symbolic fidelity to existing methods.

- The comparison to SRBench 2025 is lacking, as the most similar method in the Benchmark is EQL (2016), but there have been numerous alterations to EQL, like SINDy, EQL-div, EQL++, etc.

- The paper is lacking a complexity measure for SRBench 2025, as complexity is key comparison between neural net complexity and interpretable expressions.

**Requested Changes:**

The reviewer has broken their request into critical, analysis, and minor sections. The reviewer denotes a critical request as one needed for the paper's soundness, an analysis request as one to improve the paper's completeness, and a minor request as one to improve the paper's quality.

Critical:

- Can the authors more explicitly compare SINDy and EQL to their method? How does the DNAC layer differ from the LASSO optimization used in SINDy and EQL? How does the EQL structure compare to MSRNET or ExMSRNet's structure?

- Complexity comparison is missing for SRBenchmark, as the trade-off between equation length and accuracy is a crucial component of symbolic regression.

- Tables 3-5 in the appendix need to have the percentage of the time the equation is recovered and a description of how the authors identify if the correct equation was found. Additionally, a comparison to other symbolic regression methods is missing to support the claim "MSRNet variants achieves a balance between predictive accuracy, symbolic fidelity, and computational efficiency".

- Can the authors compare other structured neural frameworks to MSRNet in Table 2?

- Can the authors compare MSRNet to other structured neural frameworks, additional EQL-like networks, and/or SINDy in SRBench 2025?

Analysis:

- How or why does MSRNet reduce gradient starvation? Can you provide evidence that this difference is caused by gradient starvation? Such as showing that the NPU has a significantly higher error due to poorly optimized weights? Or can you provide additional experiments on a family of expressions that result in gradient starvation?

- Can the authors provide additional analysis on SRBench 2025 or additional experiments showing MSRNet and ExMSRNet performance alongside dimensionality to support the statement "his approach scales more favorably to higher-dimensional settings than other neuro-symbolic methods, while retaining the ability to recover compact symbolic expressions," compared to existing symbolic regression methods?

- Can the authors provide an analysis on why MSRNet and ExMSRNet have highly consistent performance across all SRBench 2025 fundamental equations, while existing symbolic regression methods are inconsistent?

- Can the authors provide examples of the first limitation given in Section 7, as MSRNet and ExMSRNet recover every expression in Table A?

Minor:

- Several small grammatical errors in subject-verb agreement.
- Discrepancy between Wh and kWh in the caption of Figure 8 and the x-axis.
- "ExMSRNet getting perfect test R2 on exp with every run." Table 1 shows that ExMSRNet's 3rd quartile on exp is at 0.99.
- Can the authors provide runtime comparisons in addition to the energy usage?
- Can the authors provide hyperparameter settings for all SNFs run in experiment 1?

---

> ### Author Response · Authors · 2026-06-18
> **Response to Reviewer XvDK (1/2)**
>
> __Thank you for your thoughtful and constructive feedback. We address each point below. Since there is a character limit in openreview, we have split our response in two parts.__
>
> ___
>
> > The authors admit that NMU can compare favorably to ExMSRNet in terms of accuracy, but do not provide a comparison of NMU (or other SNFs) for extracting sparse, interpretable symbolic equations.
>
> Neural Multiplicative Unit (NMU) is used to model multiplicative relations only. It cannot represent arbitrary powers including divisions. To work with NMU, we require explicit feature expansion, which increases the hidden dimension by a factor of $|W|$. This is supported by Table 1 (Synthetic benchmark comparison across methods) where NMU consumes a mean energy of $12.44 \times 10^{-3}\mathrm{Wh}$ compared to $4.52 \times 10^{-3}\mathrm{Wh}$ used by MSRNet for a single model training. For the experiments of table 1, we apply feature expansion on the output of DNAC/Linear layer having dimension 5. Therefore, the NMU is used for the hidden dimension of $5 \cdot |W|$. For simple datasets such as Lure, this is acceptable and provides competitive performance. However, for practical purposes, the NMU expansion should be applied before feature selection. On high dimensional datasets, this is not feasible and results in poor performance.
>
> ___
>
> > In Table 1, the authors show that MSRNet outperforms NPU and NALU variants on Lure. This result for MSRNet, which reduces gradient starvation, is helpful but limiting, as the sample size is a singular problem, and the authors provide no explicit theoretical guarantees for this phenomenon.
>
> We define Lure as a family of functions that contains two components: a core term and a lure term that consumes most of the training gradient. The given lure dataset is a representative of the whole family of functions. Figure 4 (Scarce-Data Regime) shows that MSRNet variants achieve high $R^2$ when a fraction of training data is used while NPU and NALU collapses, which is consistent with reduced gradient starvation. The theoretical aspects of MSRNet are intentionally kept minimal in the paper, which is consistent with the theoretical standards of closely related published work. However, we would add various theoretical justifications regarding gradient flow and entropy regularisation in MSRNet variants against unconstrained multiplicative neural networks in the appendix.
>
> ___
>
> > How does the DNAC layer differ from the LASSO optimization used in SINDy and EQL? How does the EQL structure compare to MSRNET or ExMSRNet's structure?
>
> __DNAC vs SINDy/LASSO:__ SINDy + LASSO works with a pre-expanded candidate library $\Theta(x)\in\mathbb{R}^{N\times p}$, where $p$ grows combinatorially with input dimension and polynomial degree (Section 4.3, Eq. 23). The linear regression then fits a sparse coefficient vector over this fixed expansion. In comparison, DNAC performs input selection implicitly within the neural architecture. It does NOT require pre-expansion. The weights are discrete in {−1, 0, 1} via softmax formulation. This allows for joint optimization of the whole model. In LASSO, sparsity is _encouraged_ through an $\ell_1$ penalty on continuous weights, which does NOT guarantee discrete/exact zero selection and still suffers from representational ambiguity (Section 4.1). DNAC's softmax-based discretization reduces this ambiguity.
>
> __MSRNet variants vs EQL:__ EQL (Martius and Lampert, 2016; Sahoo et al., 2018) embeds fixed symbolic operators as activation functions in a fully-connected network. Its weights are continuous (no discrete operator selection is used). As a result, EQL is susceptible to the equivalence-class problem described in Section 4.1. MSRNet replaces continuous weights with softmax distributions over a fixed rational vocabulary W. This explicitly collapses equivalence classes. Additionally, EQL does not perform feature selection whereas MSRNet's DNAC does this explicitly and in a differentiable manner.
>
> We have already compared MSRNet against other methods in Response to Reviewer VUN5. We will add a dedicated comparison paragraph in Section 2.2 explicitly discussing these distinctions.

---

> > ### Author Response · Authors · 2026-06-18
> > **Response to Reviewer XvDK (2/2)**
> >
> > > Complexity comparison is missing for SRBenchmark, as the trade-off between equation length and accuracy is a crucial component of symbolic regression.
> >
> > We believe that raw equation length is an imperfect representation of equation complexity. As an example, the Newton's law of gravitation contains a gravitational constant G. If model A approximates the constant to a certain degree of precision, and achieves competitive results, and if model B approximates the constant to a much greater degree of precision, but achieves only slightly better results, we should NOT penalise model B. Representation of equation complexity is a different topic in itself and we don't think that it should be a part of this work. A better representative of equation complexity could be equation sparsity. We would release a CSV file which contains equation (and sparsity) for each model-hyperparameter configuration. That would allow us to compare the results of MSRNet variants against other symbolic regression methods.
> >
> > ___
> >
> > > Tables 3-5 in the appendix need to have the percentage of the time the equation is recovered and a description of how the authors identify if the correct equation was found
> >
> > We have described in Section 6.5 (Taylor-Series Simulation of Exponential and Sine Functions) regarding AI Feynman datasets, "We consider an equation to be “recovered” if the model weights result in an equivalent equation. For trigonometric and exponential relations, we consider their equivalent taylor-approximation to be valid". We would add explicit column for the percentage of the times the equation is recovered in tables 3-5 of appendix.
> >
> > ___
> >
> > > Can the authors compare other structured neural frameworks to MSRNet in Table 2?
> >
> > Table 2 describes the Robustness to hidden dimension in MSRNet. Did you mean Figure 6? We have explicitly excluded other structured neural frameworks in Figure 6 (SRBench 2025 black-box track heatmap across various symbolic-regression baselines and MSRNet variants) and Figure 7 (SRBench 2025 fundamental-equation track heatmap across various symbolic-regression baselines and MSRNet variants) as they are NOT symbolic regression methods.
> >
> > If referring to table 2, We would add another table for various multiplicative neural networks (NALU, NMU, NPU, and RealNPU) comparing their robustness to hidden dimensions. Note that entropy is NOT defined for those methods.
> >
> > ___
> >
> > > Can the authors compare MSRNet to other structured neural frameworks, additional EQL-like networks, and/or SINDy in SRBench 2025?
> >
> > SINDy and EQL-like methods were NOT available as pre-configured baselines in the SRBench 2025 repository (https://github.com/cavalab/srbench/tree/srbench_2025), which provides standardized implementations for the methods included in Figures 6 and 7. Including SINDy and other EQL  variants with equivalent hyperparameter tuning and seeding would require extensive integration effort to ensure fair comparison. We will note this limitation explicitly in the revision and include a direct model comparison (NOT benchmark comparison) in Section 2.3, comparing MSRNet with EQL variants and SINDy in terms of architecture, operator selection mechanism, and computational scaling.
> >
> > ___
> >
> > > Can the authors provide examples of the first limitation given in Section 7, as MSRNet and ExMSRNet recover every expression in Table A?
> >
> > As an example, the function $f(x)=\sin(\sin x)$ requires a composition of transcendental functions that cannot be represented by any finite-degree polynomial approximation within a single MSRNet module. Similarly, $f(x)=x^{\pi}$ cannot be expressed with rational exponents from $W$. A particularly interesting case is $f(x)=abs(x)$. While EQL can explicitly represent this function, MSRNet variants represent it as $f(x)=(x^2)^\frac{1}{2}$. These are some cases where ExMSRNet's gating provides partial improvements, but does not fully resolve the limitation.
> >
> > ___
> >
> > > "ExMSRNet getting perfect test R2 on exp with every run." Table 1 shows that ExMSRNet's 3rd quartile on exp is at 0.99.
> >
> > Table 1 shows that ExMSRNet's _1st_ quartile is at $0.99$. The reason for this is that even though ExMSRNet is able to get exp on every run, the readout layer weight is NOT exactly $1.0$ in each run since there is no restriction on the last layer to have specific weights only.
> >
> > ___
> >
> > > Can the authors provide runtime comparisons in addition to the energy usage?
> >
> > Runtime comparisons vary with optimizations and parallelism. With no optimization (simple pytorch code with various CPU-GPU transfers) and no parallelisations, each model-hyperparameter configuration takes about 7-10 seconds to train over 40k training samples with GPU P100. It takes about 10-12 seconds for it to train on CPU only. We will release a CSV file which contains runtime for each model-hyperparameter configuration.

---

> > > ### Comment · Reviewer_XvDK · 2026-06-26
> > >
> > > The reviewer thanks the authors for their proposed adjustments, with the additional comparison to EQL and SINDy in section 2.3 being prominent, however the reviewer has two main concerns remaining.
> > >
> > > R1: "We believe that raw equation length is an imperfect representation of equation complexity. As an example, the Newton's law of gravitation contains a gravitational constant G. If model A approximates the constant to a certain degree of precision, and achieves competitive results, and if model B approximates the constant to a much greater degree of precision, but achieves only slightly better results, we should NOT penalise model B. Representation of equation complexity is a different topic in itself and we don't think that it should be a part of this work."
> > >
> > > C1: Sparsity is a different problem then complexity, and while complexity (model size) is not a perfect measure it can serve as a roughly analogous representation of interpretability. Symbolic regression is focused on generating interpretable expressions, and balancing the tradeoff between complexity and accuracy. The reviewer recognizes that this is not the focus of the work and would be fine with it being in the appendix, but requests it for a completeness of a symbolic regression paper.
> > >
> > > R2: We have described in Section 6.5 (Taylor-Series Simulation of Exponential and Sine Functions) regarding AI Feynman datasets, "We consider an equation to be “recovered” if the model weights result in an equivalent equation. For trigonometric and exponential relations, we consider their equivalent taylor-approximation to be valid". We would add explicit column for the percentage of the times the equation is recovered in tables 3-5 of appendix.
> > >
> > > C2: While this is helpful, it does not provide a fair comparison to existing symbolic regression methods, as EQL and SINDy networks are required to handle trigonometric and exponential functions to solve these equations. The reviewer would ask for at least one of the following is added to provide a fair comparison to other methods; average R^2 score, percent accuracy, or symbolic recovery rate without with trigonometric or exponential approximation.
> > >
> > > Lastly, the reviewer would like to clarify that they are referencing to Table 2, as the reviewer believes providing a comparison to other methods in these metrics would help improve completeness of the paper, but recognizes that it isn't not necessary given the focus of the work being applying MSRNet to Symbolic Regression.

---

> > > > ### Author Response · Authors · 2026-06-27
> > > >
> > > > __we thank the reviewer XvDK for their timely response and appreciate their effort.__
> > > >
> > > > ___
> > > >
> > > > C1: We would show the aggregate metrics (including average equation length) for each method in appendix for various datasets (from synthetic, Feynman AI, and SRBench Fundamental Equations track only).
> > > >
> > > > C2: EQL and SINDy _can_ have trigonometric functions as hyperparameters (as can MSRNet variants). We would add median $R^2$ as a column to the appendix tables and equation recovery rate with _and_ without taylor-series approximation. Note that for equations involving trigonometric equations, the equation recovery rate would be absolute 0 since MSRNet variants cannot represent trigonometric equations.

---

### Comment · Action_Editor_hijP · 2026-06-12
**Respond to Reviewers**

Dear Authors,

You will have seen that sufficient reviews have been provided. Please make sure to respond to all the reviewers' comments and questions. Then the reviewers will start discussions.

Regards, A.E.